# Convergence Analysis of Fractional Gradient Descent

**Ashwani Aggarwal**
**Department of Computer Science**
**University of California, Los Angeles**
`ashraggarwal@ucla.edu`

**Reviewed on OpenReview:** `https://openreview.net/forum?id=OycfV3Mhfq`

## Abstract

Fractional derivatives are a well-studied generalization of integer order derivatives. Naturally, for optimization, it is of interest to understand the convergence properties of gradient descent using fractional derivatives. Convergence analysis of fractional gradient descent is currently limited both in the methods analyzed and the settings analyzed. This paper aims to fill in these gaps by analyzing variations of fractional gradient descent in smooth and convex, smooth and strongly convex, and smooth and non-convex settings. First, novel bounds will be established bridging fractional and integer derivatives. Then, these bounds will be applied to the aforementioned settings to prove linear convergence for smooth and strongly convex functions and $O(1/T)$ convergence for smooth and convex functions. Additionally, we prove $O(1/T)$ convergence for smooth and non-convex functions using an extended notion of smoothness - Hölder smoothness - that is more natural for fractional derivatives. Finally, empirical results will be presented on the potential speed up of fractional gradient descent over standard gradient descent as well as some preliminary theoretical results explaining this speed up.

## 1 Introduction

Fractional derivatives (David et al., 2011), Oldham & Spanier (1974), (Luchko, 2023) as a generalization of integer order derivatives are a much studied classical field with many different variations. One natural question to ask is if they can be utilized in optimization similar to gradient descent which utilizes integer order derivatives.

To motivate the usefulness of fractional gradient descent, we can observe from experiments in Shin et al. (2021) that their Adaptive Terminal Caputo Fractional Gradient Descent (AT-CFGD) method is capable of empirically outperforming standard gradient descent in convergence rate. In addition, they showed that training neural networks based on their AT-CFGD method can give faster convergence of training loss and lower testing error. Figure 1 depicts convergence on a quadratic function for standard gradient descent as well as AT-CFGD and the method in Corollary 15 labeled Fractional Descent guided by Gradient. For specifically picked hyperparameters, both of these fractional methods can significantly outperform standard gradient descent. This suggests that study on the application of fractional derivatives to optimization has a lot of potential.

The basic concept of a fractional derivative is a combination of integer-order derivatives and fractional integrals (since there is an easy generalization for integrals through Cauchy repeated integral formula). The fractional derivative that will be studied here is the Caputo Derivative since it has nice analytic properties. The definition from Shin et al. (2021) is as follows where $\Gamma$ is the gamma function generalizing the factorial (many texts give the definition only for $x > c$, but this will be extended later on).

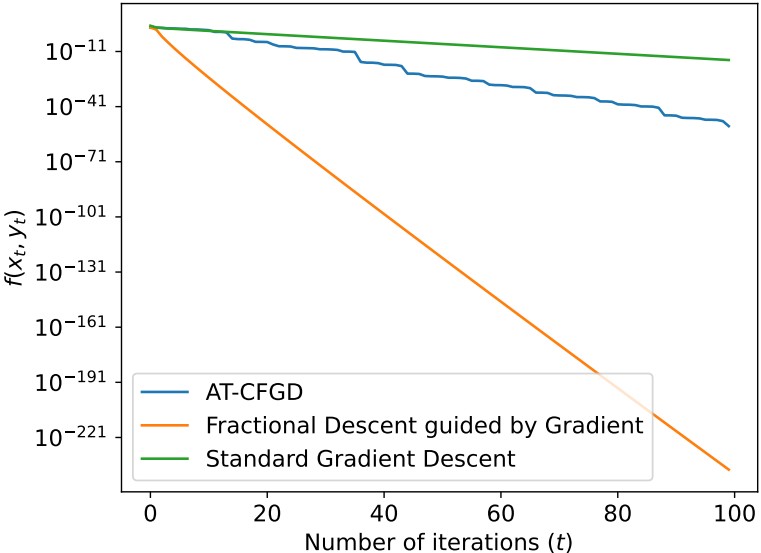

Figure 1: Convergence of descent methods on function $f(x, y) = 10x^2 + y^2$ beginning at $x = 1, y = -10$. In all cases, the optimal (not theoretical) step size is used. AT-CFGD is as described in Shin et al. (2021) with $x^{(-1)} = 1.5, y^{(-1)} = -10.5$, $\alpha = 1/2$, $\beta = -4/10$. Fractional Descent guided by Gradient is the method discussed in Corollary 15 with $\alpha = 1/2$, $\beta = -4/10$, $\lambda_t = -\frac{0.0675}{(t+1)^{0.2}}$ in $x_t - c_t = -\lambda_t \nabla f(x_t)$.

**Definition 1** (Left Caputo Derivative). For $x > c$, the (left) Caputo Derivative of $f : \mathbb{R} \to \mathbb{R}$ of order $\alpha$ is $(n = \lceil \alpha \rceil)$:

$$^{C}D_c^{\alpha+}f(x) = \frac{1}{\Gamma(n-a)} \int_c^x \frac{f^n(t)}{(x-t)^{\alpha-n+1}} dt.$$

The main contributions of this paper are highlighted as follows:

1. First, we establish novel inequalities that connect fractional derivatives to integer derivatives. This is important since properties like smoothness, convexity, and strong convexity are expressed in terms of gradients (first derivatives). Without these inequalities, assuming these properties would be meaningless from the perspective of a fractional derivative.

2. Next, we apply these inequalities to derive convergence results for smooth and strongly convex functions. In particular, the fractional gradient descent method we examine is a variation of the AT-CFGD method of Shin et al. (2021). Theorem 14 gives a linear convergence rate for this method on different hyperparameter domains for single dimensional functions. Corollary 15 extends these results to higher dimensional functions that are sums of single dimensional functions. Lastly, Theorem 16 gives linear convergence results for general higher dimensional functions.

3. Continuing onwards, we examine smooth and convex functions. In particular, if hyperparameters satisfy certain assumptions, Theorem 17 and Theorem 18 give a $O(1/T)$ convergence rate for the fractional gradient method similar to standard gradient descent.

4. The last setting we examine is smooth, but non-convex functions. For this setting, we use an extension of standard smoothness - Hölder smoothness - in which standard gradient descent needs varying learning rate to converge. We establish a general $O(1/T)$ convergence result to a stationary point in Theorem 20 and show that fractional gradient descent with well chosen hyperparameters is more natural for optimizing Hölder smooth functions.

5. Finally, we present results which show potential for fractional gradient descent speeding up convergence compared to standard gradient descent. One main point of inquiry is how fractional gradient descent,

in some cases, is able to significantly beat the theoretical worse case rates derived (which are at best as good as gradient descent). Empirically, it seems that this can be explained by the optimal learning rate far exceeding the theoretical learning rate. Another interesting example that is explored is how functions with the same amount of smoothness and strong convexity might have different preferences between fractional and standard gradient descent. Lastly, some basic theoretical results on this speed up are given for quadratic functions.

## 2 Related Work

Fractional gradient descent is an extension of gradient descent so its natural context is in the literature surrounding gradient descent. Gradient descent as an idea is classical, however, there are a number of variations some of which are very recent (Ruder, 2016). One variation is acceleration algorithms including momentum which incorporates past history into the update rule and Nesterov's accelerated gradient which improves on this by computing the gradient with look-ahead based on history. Another line of variation building on this is adaptive learning rates with algorithms including Adagrad, Adadelta, and the widely popular Adam (Kingma & Ba, 2017). There is also a descent method combining the ideas of Nesterov's accelerated gradient and Adam called Nadam.

Moving to fractional gradient descent, it is not possible to simply replace the derivative in gradient descent with a fractional derivative and expect convergence to the optimum. This is because, as discussed in Wei et al. (2020) and Wei et al. (2017), the point at which fractional gradient descent converges is highly dependent on the choice of terminal, $c$, and may not have zero gradient if $c$ is fixed. This leads to a variety of methods discussed in Wei et al. (2020) and Shin et al. (2021) to vary the terminal or order of the derivative to achieve convergence to the optimum point. Later on, the former will be done in order to guarantee convergence. Other papers take a completely different approach like Hai & Rosenfeld (2021) which opts to generalize gradient flow by changing the time derivative to a fractional derivative thus bypassing these problems.

One reason why there are so many different approaches across the literature is that fractional derivatives can be defined in many different ways (David et al., 2011) (the most commonly talked about include the Caputo derivative used in this paper as well as the Riemann-Liouville derivative). Some papers like Sheng et al. (2020) also choose simply to take a 1st degree approximation of the fractional derivative which can be expressed directly in terms of the 1st derivative. There are also further variations such as taking convex combinations of fractional and integer derivatives for the descent method like in Khan et al. (2018). Finally, there are extensions combining fractional gradient descent with one of the aforementioned variations on gradient descent. One example of this is in Shin et al. (2023) which extends the results of Shin et al. (2021) to propose a fractional Adam optimizer.

To provide further motivation for the usefulness of this field, there are many papers studying the application of fractional gradient descent methods on neural networks and other machine learning problems. For example, Han & Dong (2023) and Wang et al. (2017) have shown improved performance when training back propagation neural networks with fractional gradient descent. In addition, other papers like Wang et al. (2022) and Sheng et al. (2020) have trained convolutional neural networks and shown promising performance on the MINST dataset. Applications to further models have also been studied in works like Khan et al. (2018) which studied RBF neural networks and Tang (2023) which looked at optimizing FIR models with missing data.

In general, when reading through the literature, many fractional derivative methods have only been studied theoretically for a specific class of functions like quadratic functions in Shin et al. (2021) or lack strong convergence guarantees like in Wei et al. (2020). Detailed theoretical results like those of Hai & Rosenfeld (2021) and Wang et al. (2017) are fairly rare or limited. Thus, one main goal of this paper is to develop methodology for proving theoretical convergence results in more general smooth, convex, or strongly convex settings. As an interesting aside, fractional derivatives are generally defined by integration which means they fall under the field of optimization called nonlocal calculus which has been studied in general by Nagaraj (2021).

## 3 Relating Fractional Derivative and Integer Derivative

Before beginning the theoretical discussion, it is important to note that for the most part, all of the setup will be done in terms of single-variable functions. Although this might appear odd, due to how the fractional gradient in higher dimensions is defined, when generalizing to higher dimensional functions much of the math ends up being coordinate-wise. In fact, all of the later results will generalize to higher dimensions following very similar logic as the single dimension case.

Before presenting bounds relating fractional and integer derivatives, we need to extend the definition of the fractional derivative to $x < c$. For the extension, the definition of the right Caputo Derivative from Shin et al. (2021) is used:

**Definition 2** (Right Caputo Derivative). For $x < c$, the right Caputo Derivative of $f : \mathbb{R} \to \mathbb{R}$ of order $\alpha$ is ($n = \lceil \alpha \rceil$):

$$^C D_c^{\alpha -} f(x) = \frac{(-1)^n}{\Gamma(n - a)} \int_x^c \frac{f^n(t)}{(t - x)^{\alpha - n + 1}} dt.$$

From this, we can unify both the left and right Caputo Derivatives into one definition.

**Definition 3** (Caputo Derivative). The Caputo Derivative of $f : \mathbb{R} \to \mathbb{R}$ of order $\alpha$ is ($n = \lceil \alpha \rceil$):

$$^C D_c^{\alpha} f(x) = \frac{(\operatorname{sgn}(x - c))^{n-1}}{\Gamma(n - a)} \int_c^x \frac{f^n(t)}{|x - t|^{\alpha - n + 1}} dt$$

where sgn is the sign function.

In order to motivate calling this a fractional derivative, we can compute limits as the order of the derivative tends to an integer following the logic of 5.3.1 in Atangana (2018).

**Theorem 4.** Choose some $\alpha \in \mathbb{R}$ and let $n = \lceil \alpha \rceil$. Suppose $f : \mathbb{R} \to \mathbb{R}$ is $n$ times differentiable and $f^n(t)$ is absolutely continuous throughout the interval $[\min(x, c), \max(x, c)]$. Then,

- $\lim_{\alpha \to n} {}^C D_c^{\alpha} f(x) = \operatorname{sgn}(x - c)^n f^n(x)$,

- $\lim_{\alpha \to n-1} {}^C D_c^{\alpha} f(x) = \operatorname{sgn}(x - c)^{n-1}(f^{n-1}(x) - f^{n-1}(c))$.

*Proof.* Proof is via integration by parts, see Appendix A.1 for details. $\qquad \square$

One interesting point of this theorem is that for odd $n$, a extra $\operatorname{sgn}(x - c)$ term appears in the upper limit in addition to the ordinary $n$th derivative. This will end up motivating coefficients that are proportional to $\operatorname{sgn}(x - c)$ to cancel this term out.

Next, we present a key theorem relating the first derivative with the fractional derivative. To do so, we modify Proposition 3.1 of Hai & Rosenfeld (2021) with the extended domain of $x < c$.

**Theorem 5** (Relation between First Derivative and Fractional Derivative). Suppose $f : \mathbb{R} \to \mathbb{R}$ is continuously differentiable. Let $\alpha \in (0, 1]$. Define $\zeta_x(t)$ as:

$$\zeta_x(t) = f(t) - f(x) - f'(x)(t - x).$$

Then, we have:

$$^C D_c^{\alpha} f(x) - \frac{f'(x)(x - c)}{\Gamma(2 - \alpha)|x - c|^{\alpha}} = \frac{-\zeta_x(c)}{\Gamma(1 - \alpha)|x - c|^{\alpha}} - \frac{\alpha \operatorname{sgn}(x - c)}{\Gamma(1 - \alpha)} \int_c^x \frac{\zeta_x(t)}{|x - t|^{\alpha + 1}} dt.$$

*Proof.* Proof is via integration by parts following the logic of Proposition 3.1 of Hai & Rosenfeld (2021), see Appendix A.2 for details. $\qquad \square$

**Corollary 6.** Suppose $f : \mathbb{R} \to \mathbb{R}$ is continuously differentiable. Let $\alpha \in (0, 1]$. If $f$ is convex,

$$^{C}D_c^{\alpha} f(x) \leq \frac{f'(x)(x-c)}{\Gamma(2-\alpha)|x-c|^{\alpha}}.$$

*Proof.* Start with Theorem 5's conclusion. Convexity implies $\zeta_x(t) \geq 0$. Thus, the first term on the RHS is immediately $\leq 0$. In the second term on the RHS, $\text{sgn}(x - c)$ fixes the integral to be in the positive direction. Therefore, the second term is also $\leq 0$ since the interior of the integral is positive. $\square$

In the interest of getting the most general results possible, we need to extend the notion of $L$-smooth and $\mu$-strongly convex as will be defined here following Nesterov (2015). As will be shown in later results, this extended notion could be more natural for fractional gradient descent.

**Definition 7.** $f : \mathbb{R}^k \to \mathbb{R}$ is $(L, p)$-Hölder smooth for $p > 0$ if:

$$|f(y) - f(x) - \langle \nabla f(x), y - x \rangle| \leq \frac{L}{1+p} \|y - x\|_{1+p}^{1+p}.$$

If $p = 1$, $f$ is $L$-smooth.

Note that when convexity is assumed, the absolute value signs on the LHS do not matter since convexity means the LHS is non-negative.

**Definition 8.** $f : \mathbb{R}^k \to \mathbb{R}$ is $(\mu, p)$-uniformly convex for $p > 0$ if:

$$f(y) - f(x) - \langle \nabla f(x), y - x \rangle \geq \frac{\mu}{1+p} \|y - x\|_{1+p}^{1+p}.$$

If $p = 1$, $f$ is $\mu$-strongly convex.

Note that here the norm changes with $p$ as well. Although this may be somewhat non-standard, the linearity over dimension is useful in proving later results.

In later sections, both $p = 1$ and $p \neq 1$ cases will be studied so for this section we derive bounds in the most general $p \neq 1$ case. For $p = 1$, Zhou (2018) provides many useful properties that will be leveraged in proving convergence rates later on. These smoothness and convexity definitions are now combined with Theorem 5 to get two more useful inequalities.

**Corollary 9.** Suppose $f : \mathbb{R} \to \mathbb{R}$ is continuously differentiable. Let $\alpha \in (0, 1]$. If $f$ is $(L, p)$-Hölder smooth,

$$\left| \frac{f'(x)(x-c)}{\Gamma(2-\alpha)|x-c|^{\alpha}} - {}^{C}D_c^{\alpha} f(x) \right| \leq \frac{L}{\Gamma(1-\alpha)(1+p-\alpha)} |x-c|^{1+p-\alpha}.$$

*Proof.* Proof is a straightforward application of Theorem 5 and the definition of $(L, p)$-Hölder smooth, see Appendix A.3 for details. $\square$

**Corollary 10.** Suppose $f : \mathbb{R} \to \mathbb{R}$ is continuously differentiable. Let $\alpha \in (0, 1]$. If $f$ is $(\mu, p)$-uniformly convex,

$$\frac{f'(x)(x-c)}{\Gamma(2-\alpha)|x-c|^{\alpha}} - {}^{C}D_c^{\alpha} f(x) \geq \frac{\mu}{\Gamma(1-\alpha)(1+p-\alpha)} |x-c|^{1+p-\alpha}.$$

*Proof.* The proof is identical to that of Corollary 9 simply replacing $L$ with $\mu$ and using $\geq$ instead of $\leq$. $\square$

We now will make use of these bounds to derive convergence rates for several different settings.

# 4 Smooth and Strongly Convex Optimization

## 4.1 Fractional Gradient Descent Method

This section will focus on a modification of the AT-CFGD method from Shin et al. (2021) from the perspective of smooth and strongly convex twice differentiable functions. This study is a natural extension of prior work since they focused primarily on quadratic functions. For this section, we also assume that $p = 1$ since $p \neq 1$ introduces many complications stemming from the non-existence of inner products corresponding to $L^{1+p}$ norms if $p \neq 1$. The fractional gradient descent method is defined for $f : \mathbb{R} \to \mathbb{R}$ with $\alpha \in (0, 1)$, $\beta \in \mathbb{R}$ as:

$$x_{t+1} = x_t - \eta_t \, {}^C\delta_{c_t}^{\alpha,\beta} f(x_t)$$

where

$$
\begin{aligned}
{}^C\delta_c^{\alpha,\beta} f(x) &= \frac{1}{{}^CD_c^\alpha x}({}^CD_c^\alpha f(x) + \beta|x - c|^{C} D_c^{1+\alpha} f(x)) \\
&= \frac{{}^CD_c^\alpha f(x)\Gamma(2-\alpha)}{x - c}|x - c|^\alpha + \beta|x - c|\frac{{}^CD_c^{1+\alpha} f(x)\Gamma(2-\alpha)}{x - c}|x - c|^\alpha.
\end{aligned}
$$

The intuition for this method is given by Theorem 2.3 in Shin et al. (2021). It states that given a Taylor expansion of $f$ around $c$, ${}^C\delta_c^{\alpha,\beta} f(x)$ is the derivative of a smoothed function where the $k$th term for $k \geq 2$ is scaled by $C_{k,\alpha,\beta} = \left(\frac{\Gamma(2-\alpha)\Gamma(k)}{\Gamma(k+1-\alpha)} + \beta\frac{\Gamma(2-\alpha)\Gamma(k)}{\Gamma(k-\alpha)}\right)$. The sign of $\beta$ therefore determines the asymptotic behavior of these coefficients with respect to $k$ since the first term goes to 0 as $k \to \infty$ and the second has asymptotic rate $\beta(k - \alpha)^\alpha$ due to Wendel's double inequality (Qi & Luo, 2013).

For this method to be complete, we need to define how to choose $c_t$. We will see that a convenient choice is $x_t - c_t = -\lambda_t \nabla f(x_t)$ for well chosen $\lambda_t$. In later experiments, this method will thus be labelled as Fractional Descent guided by Gradient. This choice of $c_t$ differs from Shin et al. (2021) whose AT-CFGD method used $c_t = x_{t-m}$ for some positive integer $m$. In practice this fractional gradient descent method can be computed with Gauss-Jacobi quadrature as described in detail in Shin et al. (2021).

In order to better understand this fractional gradient descent method, we can take the limit as $\alpha \to 1$. Calculating, we arrive at ${}^C\delta_{c_t}^{\alpha,\beta} f(x_t) = f'(x_t) - \lambda_t\beta f'(x_t)f''(x_t) = (1 - \lambda_t\beta f''(x_t))f'(x_t)$. Intuitively, if $\lambda_t\beta$ is chosen properly, this approximates $f''(x_t)^{-1}f'(x_t)$ up to proportionality which corresponds to a 2nd order gradient method update. We will see in Section 7 that we can actually accomplish similar behavior on quadratic functions even with $\beta = 0$ as long as $\alpha \in (0, 1)$.

## 4.2 Single Dimensional Results

Before we can prove convergence results, we need one more inequality for bounding the $1 + \alpha$ derivative.

**Lemma 11.** Suppose $f : \mathbb{R} \to \mathbb{R}$ is twice differentiable. If $f$ is $L$-smooth and $\alpha \in (1, 2]$. Then,

$$
{}^CD_c^\alpha f(x) \leq \frac{L}{\Gamma(3-\alpha)}|x - c|^{2-\alpha}.
$$

If $f$ is $\mu$-strongly convex and $\alpha \in (1, 2]$. Then,

$$
{}^CD_c^\alpha f(x) \geq \frac{\mu}{\Gamma(3-\alpha)}|x - c|^{2-\alpha}.
$$

*Proof.* This is a direct result of the fact that $L$-smooth implies that $f''(x) \leq L$ and $\mu$-strongly convex implies that $f''(x) \geq \mu$. See Appendix B.1 for details. □

The next theorems are the primary tool that will be used for convergence results of this method.

**Theorem 12.** Suppose $f : \mathbb{R} \to \mathbb{R}$ is twice differentiable. If $f$ is $L$-smooth and $\mu$-strongly convex, $\alpha \in (0, 1]$, $\beta \geq 0$. Then,

$$|^C\delta_c^{\alpha,\beta} f(x) - f'(x) - K_1(x - c)| \leq K_2|x - c|.$$

where $K_1 = (\frac{L+\mu}{2})(\beta - \gamma)$ and $K_2 = (\frac{L-\mu}{2})(\beta + \gamma)$ with $\gamma = \frac{1-\alpha}{2-\alpha}$. Note that the above also holds if $f$ is $L$-smooth and convex if $\mu$ is set to 0.

*Proof.* Holds by applying Corollary 9, Corollary 10, and Lemma 11. See Appendix B.2 for details. □

**Theorem 13.** Suppose $f : \mathbb{R} \to \mathbb{R}$ is twice differentiable. If $f$ is $L$-smooth and $\mu$-strongly convex, $\alpha \in (0, 1]$, $\beta \leq 0$. Then,

$$|^C\delta_c^{\alpha,\beta} f(x) - f'(x) - K_1(x - c)| \leq K_2|x - c|.$$

where $K_1 = (\frac{L+\mu}{2})(\gamma_{\alpha,\beta})$ and $K_2 = (\frac{\mu-L}{2})(\gamma_{\alpha,\beta})$ with $\gamma_{\alpha,\beta} = \beta - \frac{1-\alpha}{2-\alpha}$. Note that the above also holds if $f$ is $L$-smooth and convex if $\mu$ is set to 0.

*Proof.* Holds by applying Corollary 9, Corollary 10, and Lemma 11. See Appendix B.3 for details. □

Everything is now ready for beginning discussion of convergence results. We have two cases: $\beta \geq 0$ and $\beta \leq 0$. We can treat these cases simultaneously by defining $K_1$, $K_2$ as in Theorem 12 for the former and Theorem 13 for the latter. In both these cases, $K_2 \geq 0$, however, $K_1$ can be positive or negative. For single dimensional $f$, we get the following convergence analysis theorem.

**Theorem 14.** Suppose $f : \mathbb{R} \to \mathbb{R}$ is twice differentiable, $L$-smooth, and $\mu$-strongly convex. Set $0 < \alpha < 1$ and $\beta \in \mathbb{R}$. If $\beta \geq 0$, define $K_1$, $K_2$ as in Theorem 12; if $\beta \leq 0$, define $K_1$, $K_2$ as in Theorem 13. Let the fractional gradient descent method be defined as follows.

- $x_{t+1} = x_t - \eta_t{}^C\delta_{c_t}^{\alpha,\beta} f(x_t)$

- $\eta_t = \frac{(1 - K_1\lambda_t - K_2|\lambda_t|)\phi}{(1 - K_1\lambda_t + K_2|\lambda_t|)^2 L}$ for $0 < \phi < 2$

- $x_t - c_t = -\lambda_t f'(x_t)$ with $1 - K_1\lambda_t - K_2|\lambda_t| > \epsilon > 0$

Then, this method achieves the following linear convergence rate:

$$|x_{t+1} - x^*|^2 \leq \left[1 - (2 - \phi)\phi\frac{\mu}{L}\left(\frac{1 - K_1\lambda_t - K_2|\lambda_t|}{1 - K_1\lambda_t + K_2|\lambda_t|}\right)^2\right]|x_t - x^*|^2.$$

In particular, however, this rate is at best the same as gradient descent.

*Proof.* Follows by applying Theorem 12 and Theorem 13 to bound the fractional gradient descent operator with an approximation in terms of the first derivative. Then, the proof of standard gradient descent rate for smooth and strongly convex functions can be followed with additional error terms from the approximation depending on $x_t - c_t$. This rate is at best same as gradient descent where we would expect to see a coefficient of $1 - \frac{\mu}{L}$ since $K_2|\lambda_t| \geq 0$. See Appendix B.4 for details. □

As a remark on the condition $1 - K_1\lambda_t - K_2|\lambda_t| > 0$, consider the special case $\lambda_t \geq 0$, $K_1 \geq 0$ then this condition reduces to $\lambda_t < \frac{1}{K_1+K_2}$. Similarly, for $\lambda_t \leq 0$, $K_1 \leq 0$, this condition reduces to $\lambda_t > \frac{-1}{K_2-K_1}$.

Ultimately, this proof does not give a better rate than gradient descent. In some sense, this is a limitation of the assumptions in that everything is expressed in terms of integer derivatives making it necessary to connect them with fractional derivatives. This in turn makes the bound weaker due to additional error terms scaling on $x_t - c_t$. However, this result is still useful for providing linear convergence results on a wider class of functions since Shin et al. (2021) only studied this method applied to quadratic functions.

### 4.3 Higher Dimensional Results

We can also consider $f : \mathbb{R}^k \to \mathbb{R}$ by doing all operations coordinate-wise. Following Shin et al. (2021), the natural extension for the fractional gradient descent operator for $f$ is:

$$^C\delta_c^{\alpha,\beta} f(x) = \left[ ^C\delta_{c^{(1)}}^{\alpha,\beta} f_{1,x}(x^{(1)}), ..., ^C\delta_{c^{(k)}}^{\alpha,\beta} f_{k,x}(x^{(k)}) \right].$$

Here $f_{j,x}(y) = f(x + (y - x^{(j)})e^{(j)})$ with $e^{(j)}$ the unit vector in the $j$th coordinate.

Generalizations of Theorem 14 can now be proven. We first assume that $f$ has a special form, namely that it is a sum of single dimensional functions.

**Corollary 15.** Suppose $f : \mathbb{R}^k \to \mathbb{R}$ is twice differentiable, $L$-smooth, and $\mu$-strongly convex. Assume $f$ is of form $f(x) = \sum_{i=1}^k f_i(x^{(i)})$ where $f_i : \mathbb{R} \to \mathbb{R}$. Set $0 < \alpha < 1$ and $\beta \in \mathbb{R}$. If $\beta \geq 0$, define $K_1$, $K_2$ as in Theorem 12; if $\beta \leq 0$, define $K_1$, $K_2$ as in Theorem 13. Let the fractional gradient descent method be defined as follows.

- $x_{t+1} = x_t - \eta_t {}^C\delta_{c_t}^{\alpha,\beta} f(x_t)$

- $\eta_t = \frac{(1 - K_1\lambda_t - K_2|\lambda_t|)\phi}{(1 - K_1\lambda_t + K_2|\lambda_t|)^2 L}$ for $0 < \phi < 2$

- $x_t - c_t = -\lambda_t \nabla f(x_t)$ with $1 - K_1\lambda_t - K_2|\lambda_t| > \epsilon > 0$

Then, this method achieves the following linear convergence rate:

$$\|x_{t+1} - x^*\|_2^2 \leq \left[ 1 - (2 - \phi)\phi \frac{\mu}{L} \left( \frac{1 - K_1\lambda_t - K_2|\lambda_t|}{1 - K_1\lambda_t + K_2|\lambda_t|} \right)^2 \right] \|x_t - x^*\|_2^2.$$

In particular, however, this rate is at best the same as gradient descent.

*Proof.* If $f$ is $L$-smooth, each $f_{j,x}(y)$ is $L$-smooth according to the single dimensional definition and all relevant results hold for it. The same goes for $\mu$-strong convexity. Note that taking the derivative of each $f_{j,x}(y)$ is the same as taking the $j$th partial derivative of $f$ at $x$ with the $j$th coordinate replaced by $y$. Thus, $^C\delta_c^{\alpha,\beta} f(x) = [^C\delta_{c^{(1)}}^{\alpha,\beta} f_1(x^{(1)}), ..., ^C\delta_{c^{(k)}}^{\alpha,\beta} f_k(x^{(k)})]$. Additionally, $\nabla f(x) = [f_1'(x^{(1)}), ..., f_k'(x^{(k)})]$. As such, the optimal point of $f$ in the $i$th coordinate is the optimal point of $f_i$. Therefore, Theorem 14 holds coordinate wise. This immediately gives the result. $\qquad\square$

In the more general case, there is a single term that does not immediately generalize to higher dimensions proportional to $\langle |\nabla f(x_t)|, |x_t - x^*| \rangle$ if the absolute value is taken element wise. For the single dimension case, convexity allowed us to bypass this issue, but for higher dimensions, we have to use Cauchy–Schwarz.

**Theorem 16.** Suppose $f : \mathbb{R}^k \to \mathbb{R}$ is twice differentiable, $L$-smooth, and $\mu$-strongly convex. Set $0 < \alpha < 1$ and $\beta \in \mathbb{R}$. If $\beta \geq 0$, define $K_1$, $K_2$ as in Theorem 12; if $\beta \leq 0$, define $K_1$, $K_2$ as in Theorem 13. Let the fractional gradient descent method be defined as follows.

- $x_{t+1} = x_t - \eta_t {}^C\delta_{c_t}^{\alpha,\beta} f(x_t)$

- $\eta_t = \frac{\frac{\phi}{L}(1 - K_1\lambda_t) - \frac{2K_2|\lambda_t|}{\mu}}{(1 - K_1\lambda_t + K_2|\lambda_t|)^2}$ for $0 < \phi < 2$

- $x_t - c_t = -\lambda_t \nabla f(x_t)$ with $\frac{\phi(1 - K_1\lambda_t)}{L} - \frac{2K_2|\lambda_t|}{\mu} > \epsilon > 0$.

Then, this method achieves the following linear convergence rate:

$$\|x_{t+1} - x^*\|_2^2 \leq \left[ 1 - \frac{(2 - \phi)\mu(1 - K_1\lambda_t)\left[ \frac{\phi}{L}(1 - K_1\lambda_t) - \frac{2K_2|\lambda_t|}{\mu} \right]}{(1 - K_1\lambda_t + K_2|\lambda_t|)^2} \right] \|x_t - x^*\|_2^2.$$

In particular, however, this rate is at best the same as gradient descent.

*Proof.* Follows by very similar logic to Theorem 14 except generalized to higher dimensions by taking operations coordinate-wise. The key difference as mentioned above is a single term whose bound requires more care. Similarly, the rate is at best the same as gradient descent where we expect a $1 - \frac{\mu}{L}$ coefficient since $K_2|\lambda_t| \geq 0$ See Appendix B.5 for details. $\qquad\square$

As a remark on the condition on $\lambda_t$, in the special case of $\lambda_t \geq 0$, $K_1 \geq 0$, we have:

$$\lambda_t \left( \frac{\phi K_1}{L} + \frac{2k K_2}{\mu} \right) < \frac{\phi}{L}$$

$$\implies \lambda_t < \frac{1}{K_1 + \frac{2k K_2 L}{\phi \mu}}.$$

For implementing the learning rate for the method in this section, we note that as $L \to \infty$, $\eta_t \to 0$. Since $L$ can always be increased while still satisfying smoothness, in principle the learning rate just needs to be chosen small enough same as in standard gradient descent. Similarly, for sufficiently small $\lambda_t$, the condition will always be satisfied. Of course, finding optimal hyper-parameters is important and techniques like line search can be employed to improve convergence rate.

## 5 Smooth and Convex Optimization

This section considers optimizing a $L$-smooth and convex function, $f : \mathbb{R} \to \mathbb{R}$. For similar reasons as the previous section, the case $p \neq 1$ is deferred for future work. The fractional gradient descent method for this section will be identical to the previous section. Similarly to the prior section, we split into two cases: 1) that $f$ is a sum of single dimensional functions and 2) that $f$ is a general higher dimensional function. For the first case, we have the following.

**Theorem 17.** Suppose $f : \mathbb{R}^k \to \mathbb{R}$ is twice differentiable, $L$-smooth, and convex. Assume $f$ is of form $f(x) = \sum_{i=1}^{k} f_i(x^{(i)})$ where $f_i : \mathbb{R} \to \mathbb{R}$. Set $0 < \alpha < 1$ and $\beta \in \mathbb{R}$. If $\beta \geq 0$, define $K_1$, $K_2$ as in Theorem 12; if $\beta \leq 0$, define $K_1$, $K_2$ as in Theorem 13. Let the fractional gradient descent method be defined as follows.

- $x_{t+1} = x_t - \eta^C \delta_{c_t}^{\alpha,\beta} f(x_t)$

- $\eta = \frac{1}{L} \left[ \frac{2(1 - \lambda K_1 - |\lambda| K_2)}{1 - \lambda K_1 + |\lambda| K_2)^2} - \frac{1}{1 - \lambda K_1 - |\lambda| K_2} \right]$

- $x_t - c_t = -\lambda \nabla f(x_t)$ with $1 - \lambda K_1 > \frac{\sqrt{2}+1}{\sqrt{2}-1} |\lambda| K_2$.

Then, this method achieves the following $O(1/T)$ convergence rate with $\bar{x_T}$ as the average of all $x_t$ for $1 \leq t \leq T$:

$$f(\bar{x_T}) - f(x^*) \leq \frac{L\|x_0 - x^*\|_2^2}{\left( 4 \left( \frac{1 - \lambda K_1 - |\lambda| K_2}{1 - \lambda K_1 + |\lambda| K_2} \right)^2 - 2 \right) T}.$$

In particular, however, this rate is at best the same as gradient descent.

*Proof.* By similar reasoning as Corollary 15, we can reduce to the single dimensional case. This case follows by applying Theorem 12 and Theorem 13 to bound the fractional gradient descent operator with an approximation in terms of the first derivative. Then, the proof of standard gradient descent rate for smooth and convex functions can be followed with additional error terms from the approximation depending on $x_t - c_t$. This rate is at best the same as gradient descent (which has coefficient $\frac{L}{2T}$) since $|\lambda| K_2 \geq 0$. See Appendix C.1 for details. $\qquad\square$

For the general case, just as in the previous section, there is a single term proportional to $\langle |\nabla f(x_t)|, |x_t - x^*| \rangle$ if the absolute value is taken element wise that must be dealt with more carefully. This term ends up making the method somewhat more complicated.

**Theorem 18.** Suppose $f : \mathbb{R}^k \to \mathbb{R}$ is twice differentiable, $L$-smooth, and convex. Set $0 < \alpha < 1$ and $\beta \in \mathbb{R}$. If $\beta \geq 0$, define $K_1$, $K_2$ as in Theorem 12; if $\beta \leq 0$, define $K_1$, $K_2$ as in Theorem 13. Let the fractional gradient descent method be defined as follows.

- $x_{t+1} = x_t - \eta_t {}^C\delta_{c_t}^{\alpha,\beta} f(x_t)$

- $\eta_t = \frac{1}{L} \left[ \frac{2(1-\lambda_t K_1 - |\lambda_t| K_2)}{(1-\lambda_t K_1 + |\lambda_t| K_2)^2} - \frac{1}{1-\lambda_t K_1} \right] = \frac{1}{L(1-\lambda_t K_1)} \left[ \frac{s_t^2 - 4s_t - 1}{(s_t+1)^2} \right]$

- $x_t - c_t = -\lambda_t \nabla f(x_t)$ with $1 - \lambda_t K_1 = s_t |\lambda_t| K_2$

- $\frac{(s_{t+1}+1)^2}{s_{t+1}^2 - 4s_{t+1} - 1} + \frac{2}{s_{t+1}} \leq \frac{(s_t+1)^2}{s_t^2 - 4s_t - 1}$ with $s_0 > \sqrt{5} + 2$.

Then, this method achieves the following $O(1/T)$ convergence rate with $\bar{x_T}$ as the average of all $x_t$ for $1 \leq t \leq T$:

$$ f(\bar{x_T}) - f(x^*) \leq \frac{L}{2} \left[ \frac{(s_0+1)^2}{s_0^2 - 4s_0 - 1} + \frac{2}{s_0} \right] \frac{\|x_0 - x^*\|_2^2}{T}. $$

In particular, however, this rate is at best the same as gradient descent.

*Proof.* Follows by very similar logic to Theorem 17. The key difference as mentioned above is a single term whose bound requires more care. This term ends up causing difficulties when passing to telescope sum. This motivates step-dependent $\eta_t$ and $\lambda_t$ which are determined by an underlying increasing sequence $s_t$. This rate is at best the same as standard gradient descent since $\left[ \frac{(s_0+1)^2}{s_0^2 - 4s_0 - 1} + \frac{2}{s_0} \right] \geq 1$ and gradient descent's rate has coefficient $\frac{L}{2T}$. See Appendix C.2 for details. $\qquad\square$

For implementation, this theorem requires first choosing the $s$ sequence satisfying the algebraic condition. Then, with an approximation of $L$ (which gradient descent also requires), $\lambda_t$ and $\eta_t$ can be computed following the formulas. In general, the condition on $\lambda_t$ has two solutions: one positive and one negative.

## 6 Smooth and Non-Convex Optimization

### 6.1 Fractional Gradient Descent Method

This section will focus on fractional gradient descent in a smooth and non-convex setting. This setting turns out to be the most straightforward to generalize to $p \neq 1$ and demonstrates potential for fractional gradient descent to be more natural. Berger et al. (2020) approaches this setting by varying the learning rate in gradient descent. For this section, we will adapt their proof in 3.1 to fractional gradient descent. We use a similar fractional gradient descent method as the previous sections defined as:

$$ x_{t+1} = x_t - \eta_t {}_p^C\delta_{c_t}^{\alpha} f(x_t) $$

where (for $f : \mathbb{R} \to \mathbb{R}$):

$$ {}_p^C\delta_c^{\alpha} f(x) = \frac{{}^C D_c^{\alpha} f(x) \Gamma(2-\alpha)}{x - c} |x - c|^{\alpha - p + 1}. $$

The difference from previous sections is that the second term is dropped since Lemma 11 no longer holds and the exponent of $|x - c|$ now depends on $p$. For higher dimensional $f$, we use the same extension as in the previous sections where each component follows this definition. Namely, for $f : \mathbb{R}^k \to \mathbb{R}$, the definition is

$$ {}_p^C\delta_c^{\alpha} f(x) = \left[ {}_p^C\delta_{c^{(1)}}^{\alpha} f_{1,x}(x^{(1)}), ..., {}_p^C\delta_{c^{(k)}}^{\alpha} f_{k,x}(x^{(k)}) \right]. $$

### 6.2 Convergence Results

For easing convergence analysis computation, we leverage the following Lemma.

**Lemma 19.** Suppose $f : \mathbb{R} \to \mathbb{R}$ is continuously differentiable. If $f$ is $(L, p)$-Hölder smooth, $\alpha \in (0, 1]$, then

$$\left| f'(x)|x - c|^{1-p} - {}^C_p \delta^\alpha_c f(x) \right| \leq K|x - c|$$

where $K = \frac{L(1-\alpha)}{(1+p-\alpha)}$.

*Proof.* Using Corollary 9, rearranging terms gives this bound directly. □

Now, we present a $O(1/T)$ convergence result to a stationary point as follows.

**Theorem 20.** Suppose $f : \mathbb{R}^k \to \mathbb{R}$ is continuously differentiable, $(L, p)$-Hölder smooth, and $\forall x, f(x) \geq f^*$. Set $0 < \alpha < 1$. Define $K$ as in Lemma 19. Let the fractional gradient descent method be defined as follows.

- $x_{t+1} = x_t - \eta {}^C_p \delta^\alpha_{c_t} f(x_t)$

- $\left| x_t^{(i)} - c_t^{(i)} \right| = \lambda \sqrt[p]{\left| \frac{\partial f}{\partial x^{(i)}}(x_t) \right|}$ with $0 < \lambda < \sqrt[p]{\frac{1}{K}}$

- $0 < \eta < \sqrt[p]{\frac{(1+p)(\lambda^{1-p} - K\lambda)}{L(\lambda^{1-p} + K\lambda)^{1+p}}}$

Then, this method achieves the following convergence rate:

$$\min_{0 \leq t \leq T} \|\nabla f(x_t)\|^{1+1/p}_{1+1/p} \leq \frac{f(x_0) - f^*}{(T+1)\psi}$$

where

$$\psi = \eta \left( \lambda^{1-p} - K\lambda - \frac{L}{1+p} \eta^p (\lambda^{1-p} + K\lambda)^{1+p} \right).$$

*Proof.* Follows by applying Corollary 9 to bound the fractional derivative with an approximation in terms of the first derivative. Then, the proof as aforementioned from 3.1 of Berger et al. (2020) can be followed with additional error terms from the approximation. Careful choice of $c_t$ is required in order for the degrees of various terms to allow simplification. See Appendix D.1 for details. □

The key difference in the fractional gradient descent operator from previous sections is the exponent is now $\alpha - p + 1$ instead of $\alpha$. If we choose $\alpha = p$ (assuming $0 < p < 1$), the total order of $|x - c|$ terms becomes 0 with a remaining $\text{sgn}(x - c)$. In this case, the fractional gradient descent operator is (up to proportionality and sign correction) just a fractional derivative. Theorem 20 tells us that convergence can be achieved in this case with constant learning rate with proper choice of $c_t$ which means that our fractional gradient descent step at any $t$ is directly proportional to the fractional derivative's value. In a sense, this means that the fractional derivative is natural for optimizing $f$ when setting $\alpha = p$.

## 7 Finding the Advantage of Fractional Gradient Descent

### 7.1 Experiments

We can see that there is an obvious gap between the motivation of doing better than standard gradient descent and the theoretical results. While the theoretical results are crucial guarantees on worst case rates, they currently cannot explain how fractional gradient descent can do better. Thus, this section is dedicated to experiments trying to explain this gap.

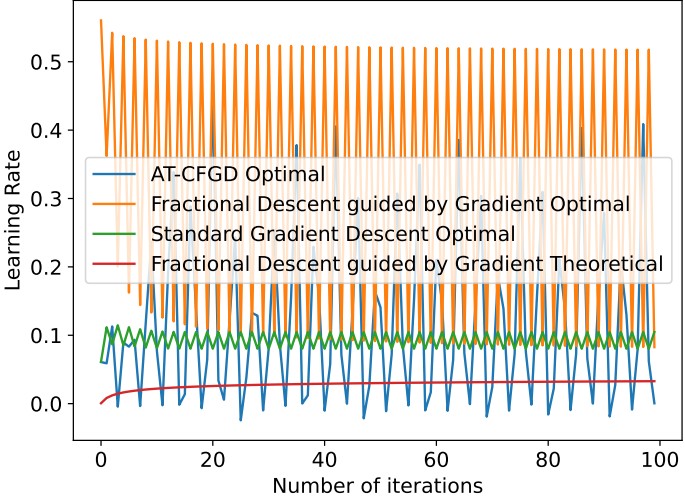

Figure 2: Learning rates used by different methods in Figure 1 with the theoretical learning rate given by Corollary 15 added.

The first thing to note is that in the experiments recorded in Figure 1, the learning rate used is not that of Corollary 15, rather it is the optimal learning rate for quadratic functions from 3.3 in Shin et al. (2021) given by:

$$\eta_t^* = \frac{\langle Ax_t + b, d_t \rangle}{\langle d_t, A d_t \rangle}.$$

for the function $\frac{1}{2} x^T A x + b^T x + c$ with descent rule $x_{t+1} = x_t - \eta_t d_t$.

We plot in Figure 2 exactly what the optimal learning rates used in Figure 1 are and how they can compare to the theoretical learning rate given by Corollary 15. The optimal learning rate for gradient descent and the theoretical learning rate from Corollary 15 tend to be significantly smaller than the optimal learning rate for fractional methods. It should be noted that the actual gradient norms may differ so the fairest comparison is between the optimal and theoretical learning rate of our method (Fractional Descent guided by Gradient).

From the equation in the discussion deriving Theorem 14: $(x_{t+1} - x^*)^2 \leq [1 - (2 - \phi)\eta_t \mu(1 - K_1 \lambda_t - K_2 |\lambda_t|)](x_t - x^*)^2$, we see that a larger learning rate directly improves the rate of convergence (assuming the larger learning rate is still valid with respect to prior assumptions). Thus, it becomes apparent that in some cases, the theoretical learning rate being much lower than necessary explains why the theoretical convergence rate is no better than that of gradient descent.

One question that could then be raised is if the current data of the assumptions is enough to be able to prove a better bound that perhaps involves a speed-up over standard gradient descent. The data with respect to a smooth and strongly convex function is two numbers - $L$, $\mu$. Other than this, the function is a black box and we would expect any bound based on these assumptions to be equivalent for any functions with the same $L$, $\mu$ assuming same hyper-parameter choices.

Looking at Figure 3 and Figure 4, we observe that despite the $L$, $\mu$ data being identical, the fractional gradient descent convergence rates are vastly different and in particular, there is no agreement over beating standard gradient descent. This suggests that to prove whether this fractional gradient descent method is better/worse than gradient descent requires more data than just $\mu$-strong convexity and $L$-smoothness about the function. Note that in order to construct this example, $\lambda_t$ had to be chosen very close to the constraint which meant that the theoretical learning rate and convergence rate are both extremely slow. This means that it may be possible to prove advantage of fractional gradient descent with tighter constraints on $\lambda_t$.

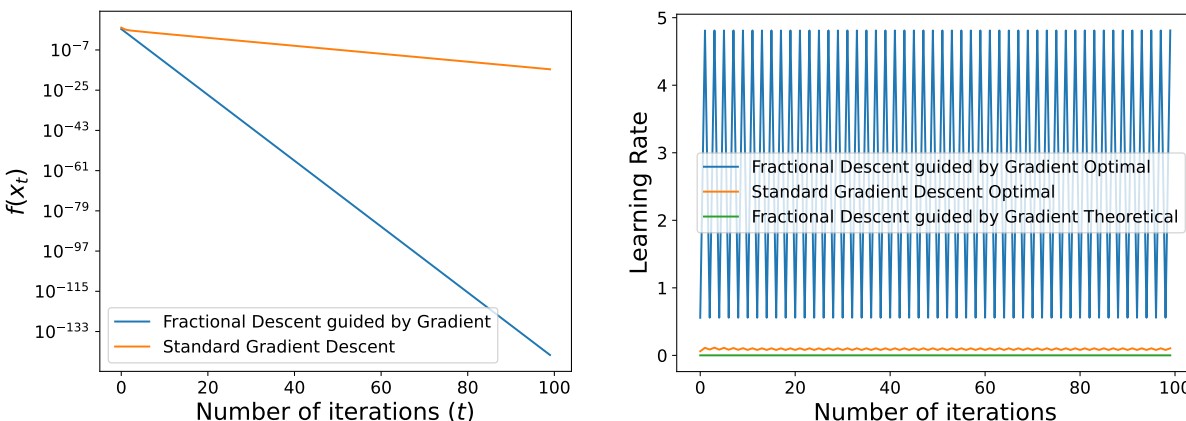

Figure 3: Comparison of fractional and standard gradient descent methods for $f(x) = x^T \text{diag}([10,1,1,1,1])x$ with $x_0 = (1,-10,5,8,-6)$. Hyper-parameters as in Corollary 15 are $\alpha = 1/2$, $\beta = -4/10$, $\lambda_t = -0.0675$

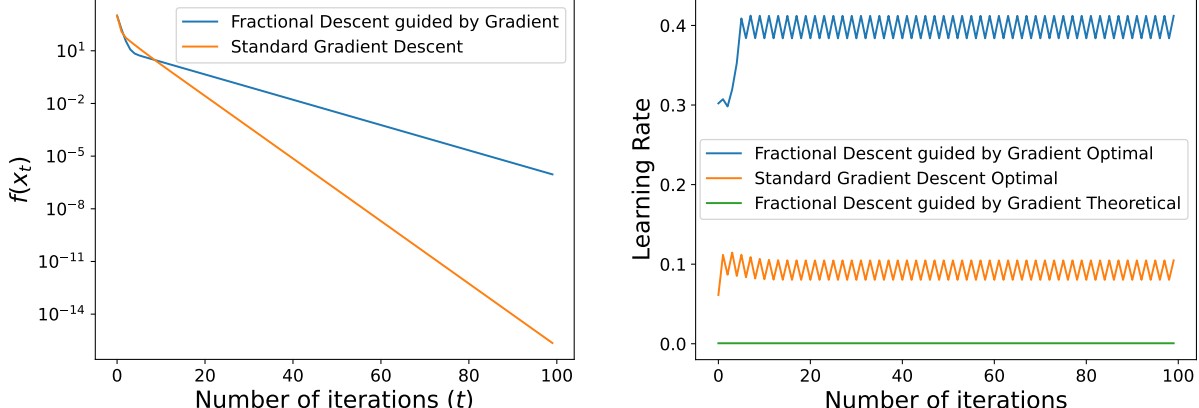

Figure 4: Comparison of fractional and standard gradient descent methods for $f(x) = x^T \text{diag}([10,1,7,9,4])x$ with $x_0 = (1,-10,5,8,-6)$. Hyper-parameters as in Corollary 15 are $\alpha = 1/2$, $\beta = -4/10$, $\lambda_t = -0.0675$

### 7.2 Quadratic Function Analysis

Motivated by these empirical findings, we present one final result applying the fractional gradient descent method from Section 4.1 to quadratic functions. This result is important in that it shows that with more information than just $L$, $\mu$, we are able to understand when the fractional gradient descent method will outperform gradient descent (at least in this limited case).

**Theorem 21.** Suppose $f(x) = \frac{1}{2}x^T \mathbf{A}x + b^T x + y_0$ for some positive definite $k \times k$ $\mathbf{A}$ with elements $a_{ij}$. Let $\mu$ be its smallest eigenvalue and $L$ be its largest eigenvalue. Choose $\beta \le 0$, $\alpha \in (0,1)$, and $\Delta \in \mathbb{R}$. Choose $c$ from $x$ so that $x - c = -\lambda \nabla f(x)$ with $\lambda = \frac{\Delta}{(\beta-\gamma)L}$ where $\gamma = \frac{1-\alpha}{2-\alpha}$. Then, we have the following:

1. Let $\mathbf{D} = \mathbf{I} - \frac{\Delta}{L}\text{diag}(\mathbf{A})$, $\mathbf{A}' = \mathbf{D}\mathbf{A}$, and $b' = \mathbf{D}b$. Then, ${}^C\delta_c^{\alpha,\beta}f(x) = \mathbf{A}'x - b'$. In particular, if $\mathbf{A}'$ is non-symmetric, this fractional gradient operator is not given by the gradient of any function.

2. Suppose that $\forall y \in \mathbb{R}^k$, we have that $\mu'\|y\|_2^2 \le y^T \mathbf{A}'y \le L'\|y\|_2^2$ and suppose that $\Delta$ is chosen so that all elements of $\mathbf{D}$ are positive. Then, we have the following linear convergence rate for fractional gradient descent:

$$\|x_t - x^*\|_2^2 \le \left[1 - \frac{\mu'}{L'}\right]^t \|x_0 - x^*\|_2^2.$$

In particular, if the condition number of $\mathbf{A}'$, $\kappa(\mathbf{A}') = \frac{L'}{\mu'}$, is smaller than $\kappa(\mathbf{A}) = \frac{L}{\mu}$, then fractional gradient descent achieves a faster convergence rate (in number of iterations).

*Proof.* The first point follows from a straightforward calculation of the fractional gradient descent operator on quadratic functions. The second point follows from a proof almost identical to that of gradient descent on quadratic functions. The linear rate of convergence for gradient descent is $1 - \frac{\mu}{L}$ so if $\kappa(\mathbf{A}') < \kappa(\mathbf{A})$, the convergence will be faster. See Appendix E.1 for more details. □

For one simple application of this theorem, consider the case where $\mathbf{A}$ is diagonal. Then $\mathbf{A}'$ is also diagonal with elements of the form $d_i = a_{ii}(1 - a_{ii}\frac{\Delta}{L})$ on the diagonal. This case corresponds to all of the above experiments. We note that $\mu'$ and $L'$ in this case are simply the smallest and largest eigenvalues of $\mathbf{A}'$ since $\mathbf{A}'$ is positive definite. Some $\mu = a_{ii}$ will be the smallest eigenvalue of $\mathbf{A}$ and some $L = a_{jj}$ will be the largest eigenvalue. In $\mathbf{A}'$, these elements become $\mu(1 - \mu\frac{\Delta}{L})$, $L(1 - \Delta)$ with $0 < \Delta < 1$. We then have the following:

$$\frac{L(1-\Delta)}{\mu(1-\mu\frac{\Delta}{L})} = \frac{L}{\mu}\frac{1-\Delta}{1-\mu\frac{\Delta}{L}} = \frac{L}{\mu}\frac{L-L\Delta}{L-\mu\Delta} \le \frac{L}{\mu} = \kappa(\mathbf{A}).$$

This appears to contradict Figure 4 since this seems to imply that fractional gradient descent should always outperform gradient descent on these simple quadratic functions. However, the key is that the LHS is not necessarily equal to $\kappa(\mathbf{A}')$. In particular, $d_i$ can also amplify $\mu < a_{ii} < L$ such that these components become the maximum eigenvalues of $\mathbf{A}'$ and can shrink components to become minimum eigenvalues. To avoid this, observe that if $\Delta \le \frac{1}{2}$, the maximum of the $d_i$ will always be where $a_{ii} = L$ and the minimum will always be where $a_{ii} = \mu$. Thus, if $\Delta \le \frac{1}{2}$, for these simple functions, fractional gradient descent always does at least as good as gradient descent. Figure 4 corresponds to the case of $\Delta = 0.99$ which does not satisfy this condition.

One more point to note is that in the limit as $\alpha \to 1$, the theorem still holds as long as $\beta \ne 0$. In particular, as long as $\beta - \gamma \ne 0$ the theorem's convergence rate has no dependence on $\beta$ and $\alpha$ since $\lambda$ inversely scaling on $\beta - \gamma$ is effectively cancelling it out. This means that in some sense if we set $\beta = 0$, we are getting the benefits of a second order method where $\alpha = 1$, $\beta \ne 0$ (see Section 4.1 for details on this limit) without actually using more than the first derivative.

For more complicated $\mathbf{A}$, applying this theorem requires comparing the condition number of $\mathbf{A}'$ against that of $\mathbf{A}$ with no easy relationship like in the previous simple case. While this theorem is limited to quadratic functions, it is promising in providing a theoretical explanation of how fractional gradient descent can outperform gradient descent - a question that prior work left open.

# 8 Future Directions

Going off of the preceding discussion, one important future direction is to search for additional assumptions to classify when this fractional gradient descent method will outperform gradient descent on general classes of functions. It is also important to develop lower bounds to prove when general fractional gradient descent methods are capable of achieving superior performance to gradient descent and to verify whether the convergence rates developed in the paper are tight.

With respect to the tightness of the bounds, one potential area for further improvement is to better handle the second term of ${}^C\delta_c^{\alpha,\beta}f(x)$ involving a fractional derivative of order between 1 and 2 with coefficient scaling with $\beta$. Currently Lemma 11 handles bounding this term, however, it is rather weak in that it loses any information that would make this term useful for convergence. This is reflected in how sending $\beta \to 0$ tends to improve the theoretical rate. Even in Theorem 21, $\beta$ could be set to 0 without any loss in the results. If we consider Theorem 4, we have for $1 < \alpha < 2$ that $\lim_{\alpha \to 1} {}^C D_c^\alpha f(x) = \text{sgn}(x - c)(f'(x) - f'(c))$. If $c$ is chosen properly, this could potentially work similarly to acceleration algorithms in gradient descent. More work is needed to understand this term outside the limit case since attempting to use integration by parts to express it in terms of the gradient fails. It is important in future work to understand if this term can be useful for proving better rates.

Another interesting direction would be to investigate the effects of changing $c_t$. Both Shin et al. (2021) and this paper use different methods of choosing $c_t$ and it is not clear which is better since it is difficult to directly compare without more theoretical results. In addition, future work could look at applying similar strategies of relating fractional and integer derivatives to different underlying fractional derivatives such as the Reimann-Liouville derivative.

One important future direction is to show convergence results for $p \neq 1$ for more settings. In this paper, we only discuss $p \neq 1$ for smooth and non-convex functions while leaving the other two settings for future work. As aforementioned the difficulties of this setting stem from the norm not being induced by an inner product on $L^{1+p}$ spaces making it difficult to expand terms of the form $\|x - y\|_{1+p}^{1+p}$ or even just $|x - y|^{1+p}$ for some arbitrary $x, y$.

Another line of thought is to bypass the need for inequalities relating fractional and integer derivatives by using convexity and smoothness definitions that only involve fractional derivatives. This would be ideal since the current proof methodology loses a lot of information about the fractional derivative in connecting it to the gradient causing the convergence rates to be bounded by those of gradient descent. One direction that may be promising is using fractional Taylor series like in Usero (2008) to construct these definitions. However, these series for Caputo Derivatives are somewhat limited in how they need to be centered at the terminal point of the derivative. If we ignore the terminal point and try to use simple assumptions like Lipschitz continuity of the fractional derivative, bad behavior around the terminal point (where the fractional derivative rapidly tends to 0) prevents it from being applicable even to simple test functions like $|x|^{1+\alpha}$. One potential solution would be to set the terminal point depending on $x$ only and not the iteration number and consider Lipschitz continuity properties after integrating this dependency. Following this line of thought, we could take the conditions from point 2 of Theorem 21 and replace $\mathbf{A}'$ with the total derivative of the fractional gradient descent operator. This could serve as a natural generalization of smoothness and strong convexity.

In conclusion, this paper proves convergence results for fractional gradient descent in smooth and strongly convex, smooth and convex, smooth and non-convex, and quadratic settings. Future work is needed in extending these results to other classes of functions and other methods to show a general guaranteed benefit over gradient descent.

### Acknowledgments

I thank Prof. Quanquan Gu for his constructive comments on this manuscript.

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

## A  Missing Proofs in Section 3 Relating Fractional Derivative and Integer Derivative

### A.1  Proof of Theorem 4

*Proof.*

$$
\begin{aligned}
{}^{C}D_c^\alpha f(x) &= \frac{\operatorname{sgn}(x-c)^{n-1}}{\Gamma(n-\alpha)} \int_c^x \frac{f^n(t)}{|x-t|^{\alpha-n+1}} dt \\
&= \frac{\operatorname{sgn}(x-c)^{n-1}}{\Gamma(n-\alpha)} \left[ -\frac{f^n(t)}{n-\alpha}|x-t|^{n-\alpha}\operatorname{sgn}(x-c)\Big|_{t=c}^{x} + \int_c^x \frac{f^{n+1}(t)|x-t|^{n-\alpha}}{n-\alpha}\operatorname{sgn}(x-c)dt \right] \\
&= \frac{\operatorname{sgn}(x-c)^n}{\Gamma(n-\alpha+1)} \left[ f^n(c)|x-c|^{n-\alpha} + \int_c^x f^{n+1}(t)|x-t|^{n-\alpha}dt \right].
\end{aligned}
$$

As $\alpha \to n$, this simplifies to:

$$
{}^{C}D_c^\alpha f(x) = \operatorname{sgn}(x-c)^n (f^n(c) + f^n(x) - f^n(c) = \operatorname{sgn}(x-c)^n f^n(x).
$$

As $\alpha \to n-1$, directly from the definition,

$$
{}^{C}D_c^\alpha f(x) = \operatorname{sgn}(x-c)^{n-1} \int_c^x f^n(t)dt = \operatorname{sgn}(x-c)^{n-1}(f^{n-1}(x) - f^{n-1}(c)).
$$

$\square$

### A.2  Proof of Theorem 5

*Proof.* First, note that for $\alpha \in (0,1]$, ${}^{C}D_c^\alpha x = \frac{x-c}{\Gamma(2-\alpha)|x-c|^\alpha}$. One interesting thing here is that this fractional derivative can be both positive and negative unlike the first derivative of a line. Also note that $d\zeta_x(t) = (f'(t) - f'(x))dt$. Therefore, we begin with the following expression:

$$
\begin{aligned}
{}^{C}D_c^\alpha f(x) - f'(x){}^{C}D_c^\alpha(x) &= \frac{1}{\Gamma(1-\alpha)} \int_c^x |x-t|^{-\alpha}(f'(t) - f'(x))dt \\
&= \frac{1}{\Gamma(1-\alpha)} \int_c^x |x-t|^{-\alpha}d\zeta_x(t) \\
&= \frac{|x-t|^{-\alpha}\zeta_x(t)}{\Gamma(1-\alpha)}\Big|_{t=c}^{x} - \frac{\alpha}{\Gamma(1-\alpha)} \int_c^x |x-t|^{-\alpha-1}\operatorname{sgn}(x-t)\zeta_x(t)dt \\
&= \frac{\zeta_x(t)}{\Gamma(1-\alpha)|x-t|^\alpha}\Big|_{t=c}^{x} - \frac{\alpha\operatorname{sgn}(x-c)}{\Gamma(1-\alpha)} \int_c^x \frac{\zeta_x(t)}{|x-t|^{\alpha+1}}dt.
\end{aligned}
$$

It remains to show that in the first term vanishes as $t \to x$ which is done using L'Hopital's Rule:

$$
\lim_{t \to x} \frac{\zeta_x(t)}{\Gamma(1-\alpha)|x-t|^\alpha} = \lim_{t \to x} \frac{f'(t) - f'(x)}{\alpha\Gamma(1-\alpha)|x-t|^{\alpha-1}\operatorname{sgn}(x-t)} = 0.
$$

The last equality is since $\alpha \in (0,1]$ so $\alpha - 1 \le 0$. $\square$

### A.3  Proof of Corollary 9

*Proof.* Note that $(L, p)$-Hölder smooth implies that $\zeta_x(t) \leq \frac{L}{1+p}|x - t|^{1+p}$. Also $1 + p - \alpha > 0$ since $p > 0, \alpha \in (0, 1]$. Thus,

$$
\begin{aligned}
\frac{f'(x)(x - c)}{\Gamma(2 - \alpha)|x - c|^\alpha} - {}^C D_c^\alpha f(x) &= \frac{\zeta_x(c)}{\Gamma(1 - \alpha)|x - c|^\alpha} + \frac{\alpha \operatorname{sgn}(x - c)}{\Gamma(1 - \alpha)} \int_c^x \frac{\zeta_x(t)}{|x - t|^{\alpha+1}} dt \\
&\leq \frac{L|x - c|^{1+p-\alpha}}{(1 + p)\Gamma(1 - \alpha)} + \frac{\alpha L \operatorname{sgn}(x - c)}{(1 + p)\Gamma(1 - \alpha)} \int_c^x |x - t|^{p-\alpha} dt \\
&\leq \frac{L|x - c|^{1+p-\alpha}}{(1 + p)\Gamma(1 - \alpha)} + \frac{\alpha L \operatorname{sgn}(x - c)}{(1 + p)\Gamma(1 - \alpha)} \operatorname{sgn}(x - c) \frac{|x - c|^{1+p-\alpha}}{1 + p - \alpha} \\
&= \frac{L}{(1 + p)\Gamma(1 - \alpha)}|x - c|^{1+p-\alpha}(1 + \frac{\alpha}{1 + p - \alpha}) \\
&= \frac{L}{\Gamma(1 - \alpha)(1 + p - \alpha)}|x - c|^{1+p-\alpha}.
\end{aligned}
$$

The other direction of the inequality follows by the same logic using instead $\zeta_x(t) \geq \frac{-L}{1+p}|x - t|^{1+p}$ and using $\geq$ instead of $\leq$.  $\square$

## B   Missing Proofs in Section 4 Smooth and Strongly Convex Optimization

### B.1   Proof of Lemma 11

*Proof.* $L$-smooth implies that $f''(x) \leq L$. Since $\alpha \in (1, 2]$,

$$
\begin{aligned}
{}^C D_c^\alpha f(x) &= \frac{\operatorname{sgn}(x - c)}{\Gamma(2 - \alpha)} \int_c^x \frac{f''(t)}{|x - t|^{\alpha-1}} dt \\
&\leq \frac{\operatorname{sgn}(x - c)}{\Gamma(2 - \alpha)} \int_c^x \frac{L}{|x - t|^{\alpha-1}} dt \\
&= \frac{\operatorname{sgn}(x - c)}{\Gamma(2 - \alpha)} \frac{|x - c|^{2-\alpha}}{2 - \alpha} \operatorname{sgn}(x - c) L \\
&= \frac{L}{\Gamma(3 - \alpha)}|x - c|^{2-\alpha}.
\end{aligned}
$$

The bound holds since the integral is in the positive direction due to $\operatorname{sgn}(x - c)$. The proof for $\mu$-strongly convex is identical except using $f''(x) \geq \mu$.  $\square$

### B.2   Proof of Theorem 12

*Proof.* We begin by upper bounding ${}^C \delta_c^{\alpha,\beta} f(x)$. Note that since both terms in it have an $(x - c)$ in the denominator, $\operatorname{sgn}(x - c)$ determines which inequality must be used. Let $R$ denote $ReLU$. Then,

$$
\begin{aligned}
{}^C \delta_c^{\alpha,\beta} f(x) &\leq f'(x) - \mu \frac{1 - \alpha}{2 - \alpha} R(x - c) + L \frac{1 - \alpha}{2 - \alpha} R(c - x) + L\beta R(x - c) - \mu\beta R(c - x) \\
&= f'(x) - \mu\gamma R(x - c) + L\gamma R(c - x) + L\beta R(x - c) - \mu\beta R(c - x) \\
&= f'(x) + (L\beta - \mu\gamma) R(x - c) + (L\gamma - \mu\beta) R(c - x).
\end{aligned}
$$

The first 3 terms on the RHS come from bounding the first term of ${}^C \delta_c^{\alpha,\beta} f(x)$ and the latter two terms come from bounding the secon term of ${}^C \delta_c^{\alpha,\beta} f(x)$. Similarly, we find that

$$
{}^C \delta_c^{\alpha,\beta} f(x) \geq f'(x) + (\mu\beta - L\gamma) R(x - c) + (\mu\gamma - L\beta) R(c - x).
$$

Observe that

$$\frac{(L\beta - \mu\gamma) - (\mu\beta - L\gamma)}{2} = \frac{(L-\mu)\beta + (L-\mu)\gamma}{2} = \frac{(L-\mu)}{2}(\beta + \gamma),$$

$$\frac{(L\gamma - \mu\beta) - (\mu\gamma - L\beta)}{2} = \frac{(L-\mu)}{2}(\beta + \gamma),$$

$$\frac{(L\beta - \mu\gamma) + (\mu\beta - L\gamma)}{2} = \frac{(L+\mu)\beta - (L+\mu)\gamma}{2} = \frac{(L+\mu)}{2}(\beta - \gamma),$$

$$\frac{(L\gamma - \mu\beta) + (\mu\gamma - L\beta)}{2} = -\frac{(L+\mu)}{2}(\beta - \gamma).$$

Using these equations gives

$$
\begin{aligned}
{}^C\delta_c^{\alpha,\beta}f(x) &\leq f'(x) + \frac{(L-\mu)}{2}(\beta + \gamma)R(x-c) + \frac{(L+\mu)}{2}(\beta - \gamma)R(x-c) \\
&\quad + \frac{(L-\mu)}{2}(\beta + \gamma)R(c-x) - \frac{(L+\mu)}{2}(\beta - \gamma)R(c-x) \\
&= f'(x) + \frac{(L+\mu)}{2}(\beta - \gamma)(x-c) + \frac{(L-\mu)}{2}(\beta + \gamma)|x-c| \\
&= f'(x) + K_1(x-c) + K_2|x-c|.
\end{aligned}
$$

Similarly, the lower bound is

$$
{}^C\delta_c^{\alpha,\beta}f(x) \geq f'(x) + K_1(x-c) - K_2|x-c|.
$$

Putting both of these bounds together gives the desired result:

$$
-K_2|x-c| \leq {}^C\delta_c^{\alpha,\beta}f(x) - f'(x) - K_1(x-c) \leq K_2|x-c|.
$$

$\square$

### B.3 Proof of Theorem 13

*Proof.* The proof begins similarly as in Theorem 12, except $\beta$ determines the sign as well.

$$
\begin{aligned}
{}^C\delta_c^{\alpha,\beta}f(x) &\leq f'(x) - \mu\frac{1-\alpha}{2-\alpha}R(x-c) + L\frac{1-\alpha}{2-\alpha}R(c-x) + LR(\beta(x-c)) - \mu R(\beta(c-x)) \\
&= f'(x) - \mu\frac{1-\alpha}{2-\alpha}R(x-c) + L\frac{1-\alpha}{2-\alpha}R(c-x) - L\beta R(c-x) + \mu\beta R(x-c) \\
&= f'(x) + (\mu\gamma_{\alpha,\beta})R(x-c) - (L\gamma_{\alpha,\beta})R(c-x) \\
&= f'(x) + K_1(x-c) + K_2|x-c|.
\end{aligned}
$$

Similarly, we find that

$$
\begin{aligned}
{}^C\delta_c^{\alpha,\beta}f(x) &\geq f'(x) + (L\gamma_{\alpha,\beta})R(x-c) - (\mu\gamma_{\alpha,\beta})R(c-x) \\
&= f'(x) + K_1(x-c) - K_2|x-c|.
\end{aligned}
$$

$\square$

### B.4 Proof of Theorem 14

*Proof.* We start with 3 point identity. Note this is where the case $p \neq 1$ breaks down since the $L^{1+p}$ norm is not induced by an inner product if $p \neq 1$.

$$
\begin{aligned}
(x_{t+1} - x^*)^2 &= (x_{t+1} - x_t)^2 + 2(x_{t+1} - x_t)(x_t - x^*) + (x_t - x^*)^2 \\
&= (\eta_t{}^C\delta_{c_t}^{\alpha,\beta}f(x_t))^2 - 2\eta_t{}^C\delta_{c_t}^{\alpha,\beta}f(x_t)(x_t - x^*) + (x_t - x^*)^2.
\end{aligned}
$$

We begin by bounding the first term:

$$
\begin{aligned}
(^C\delta_{c_t}^{\alpha,\beta}f(x_t))^2 &= ((^C\delta_{c_t}^{\alpha,\beta}f(x_t) - f'(x_t) - K_1(x_t - c_t)) + (f'(x_t) + K_1(x_t - c_t)))^2 \\
&\leq K_2^2(x_t - c_t)^2 + 2K_2|x_t - c_t||f'(x_t) + K_1(x_t - c_t)| \\
&\quad + (f'(x_t) + K_1(x_t - c_t))^2.
\end{aligned}
$$

One observation here is that we would like everything to be in terms of $f'(x_t)^2$ to make canceling more convenient later. For this purpose, choose $x_t - c_t = -\lambda_t f'(x_t)$. Thus, we get

$$
\begin{aligned}
(^C\delta_{c_t}^{\alpha,\beta}f(x_t))^2 &\leq K_2^2(\lambda_t)^2(f'(x_t))^2 + 2K_2|\lambda_t||1 - K_1\lambda_t|(f'(x_t))^2 + (1 - K_1\lambda_t)^2(f'(x_t))^2 \\
&= (K_2|\lambda_t| + |1 - \lambda_t K_1|)^2(f'(x_t))^2.
\end{aligned}
$$

Now, choose some $\phi \in (0,2)$. We now bound the second term as follows (note we assume here $\eta_t \geq 0$ since unlike the past section this makes sense):

$$
\begin{aligned}
-2\eta_t{}^C\delta_{c_t}^{\alpha,\beta}f(x_t)(x_t - x^*) &\leq 2\eta_t K_2|x_t - c_t||x_t - x^*| - 2\eta_t f'(x_t)(x_t - x^*) \\
&\quad - 2\eta_t K_1(x_t - c_t)(x_t - x^*) \\
&= 2\eta_t K_2|\lambda_t||f'(x_t)||x_t - x^*| - 2\eta_t f'(x_t)(1 - \lambda_t K_1)(x_t - x^*) \\
&= 2\eta_t K_2|\lambda_t|(f'(x_t))(x_t - x^*) - 2\eta_t f'(x_t)(1 - \lambda_t K_1)(x_t - x^*) \\
&= -2\eta_t f'(x_t)(x_t - x^*)(1 - K_1\lambda_t - K_2|\lambda_t|) \\
&\leq -\frac{\phi\eta_t}{L}(1 - K_1\lambda_t - K_2|\lambda_t|)f'(x_t)^2 \\
&\quad - (2 - \phi)\eta_t\mu(1 - K_1\lambda_t - K_2|\lambda_t|)(x_t - x^*)^2.
\end{aligned}
$$

We can drop the absolute value signs due to the convexity assumption (note this works specifically for single dimension $f$). We need both terms of the RHS to be negative for proving convergence which puts a condition on $\lambda_t$:

$$
1 - K_1\lambda_t - K_2|\lambda_t| > 0.
$$

This condition gives that $1 - K_1\lambda_t > 0$. Putting everything together gives:

$$
\begin{aligned}
(x_{t+1} - x^*)^2 &\leq \eta_t^2((K_2|\lambda_t| + 1 - \lambda_t K_1)^2(f'(x_t))^2 - \frac{\phi\eta_t}{L}(1 - K_1\lambda_t - K_2|\lambda_t|)f'(x_t)^2 \\
&\quad - (2 - \phi)\eta_t\mu(1 - K_1\lambda_t - K_2|\lambda_t|)(x_t - x^*)^2 + (x_t - x^*)^2.
\end{aligned}
$$

Now, we can figure out the learning rate since we want the first term on the RHS to be dominated by the second term.

$$
\begin{aligned}
\eta_t^2(K_2|\lambda_t| + 1 - \lambda_t K_1)^2 &\leq \frac{\phi\eta_t}{L}(1 - K_1\lambda_t - K_2|\lambda_t|) \\
\implies \eta_t &= \frac{(1 - K_1\lambda_t - K_2|\lambda_t|)\phi}{(1 - K_1\lambda_t + K_2|\lambda_t|)^2 L}.
\end{aligned}
$$

Finally, this leads to a convergence rate as follows:

$$
\begin{aligned}
(x_{t+1} - x^*)^2 &\leq [1 - (2 - \phi)\eta_t\mu(1 - K_1\lambda_t - K_2|\lambda_t|)] (x_0 - x^*)^2 \\
&= \left[1 - (2 - \phi)\phi\frac{\mu}{L}\left(\frac{1 - K_1\lambda_t - K_2|\lambda_t|}{1 - K_1\lambda_t + K_2|\lambda_t|}\right)^2\right](x_t - x^*)^2.
\end{aligned}
$$

This is a linear rate of convergence since $\epsilon$ guarantees that this equation is a contraction as $t \to \infty$. The rate is at best as good as gradient descent since $K_2|\lambda_t| \geq 0$. □

## B.5 Proof of Theorem 16

*Proof.* We follow the discussion deriving Theorem 14. We start with 3 point identity:

$$\|x_{t+1} - x^*\|_2^2 = \|x_{t+1} - x_t\|_2^2 + 2\langle x_{t+1} - x_t, x_t - x^* \rangle + \|x_t - x^*\|_2^2$$
$$= \eta_t^2 \|{}^C \delta_{c_t}^{\alpha,\beta} f(x_t)\|_2^2 - 2\eta_t \langle {}^C \delta_{c_t}^{\alpha,\beta} f(x_t), x_t - x^* \rangle + \|x_t - x^*\|_2^2.$$

For bounding the first term, this can be done coordinate wise.

$$\|{}^C \delta_{c_t}^{\alpha,\beta} f(x_t)\|_2^2 = \sum_{i=1}^k ({}^C \delta_{c_t^{(i)}}^{\alpha,\beta} f_{i,x_t}(x_t^{(i)}))^2$$
$$\leq (K_2|\lambda_t| + |1 - \lambda_t K_1|)^2 \|\nabla f(x_t)\|_2^2.$$

For bounding the second term (note $|\cdot|$ is taken element-wise, $\eta_t \geq 0$ is assumed):

$$-2\eta_t \langle {}^C \delta_{c_t}^{\alpha,\beta} f(x_t), (x_t - x^*) \rangle \leq 2\eta_t K_2 |\lambda_t| \langle |\nabla f(x_t)|, |x_t - x^*| \rangle$$
$$- 2\eta_t (1 - \lambda_t K_1) \langle \nabla f(x_t), x_t - x^* \rangle$$
$$\leq 2\eta_t K_2 |\lambda_t| [\|\nabla f(x_t)\|_2 \|x_t - x^*\|_2]$$
$$- 2\eta_t (1 - \lambda_t K_1) \langle \nabla f(x_t), x_t - x^* \rangle$$
$$\leq \frac{2\eta_t K_2}{\mu} |\lambda_t| \|\nabla f(x_t)\|_2^2 - 2\eta_t (1 - \lambda_t K_1) \langle \nabla f(x_t), x_t - x^* \rangle.$$

We see that $1 - \lambda_t K_1 > 0$ is necessary for proving convergence since we need the latter term to be negative to prove convergence. We bound the second term like in the single dimensional case as:

$$-2\eta_t (1 - \lambda_t K_1) \langle \nabla f(x_t), x_t - x^* \rangle \leq -\frac{\phi \eta_t}{L} (1 - \lambda_t K_1) \|\nabla f(x_t)\|_2^2$$
$$- (2 - \phi) \eta_t \mu (1 - \lambda_t K_1) \|x_t - x^*\|_2^2.$$

Gathering all terms yields:

$$\|x_{t+1} - x^*\|_2^2 \leq \eta_t^2 (K_2|\lambda_t| + 1 - \lambda_t K_1)^2 \|\nabla f(x_t)\|_2^2 + \frac{2\eta_t K_2}{\mu} |\lambda_t| \|\nabla f(x_t)\|_2^2$$
$$- \frac{\phi \eta_t}{L} (1 - \lambda_t K_1) \|\nabla f(x_t)\|_2^2 - (2 - \phi) \eta_t \mu (1 - \lambda_t K_1) \|x_t - x^*\|_2^2$$
$$+ \|x_t - x^*\|_2^2.$$

For the 3rd term on the RHS to dominate the first two terms, the learning rate is:

$$\eta_t^2 (K_2|\lambda_t| + 1 - \lambda_t K_1)^2 + \frac{2\eta_t K_2}{\mu} |\lambda_t| \leq \frac{\phi \eta_t}{L} (1 - \lambda_t K_1)$$
$$\implies \eta_t (K_2|\lambda_t| + 1 - \lambda_t K_1)^2 \leq \frac{\phi}{L} (1 - \lambda_t K_1) - \frac{2K_2|\lambda_t|}{\mu}$$
$$\implies \eta_t = \frac{\frac{\phi}{L}(1 - K_1 \lambda_t) - \frac{2K_2|\lambda_t|}{\mu}}{(1 - K_1 \lambda_t + K_2|\lambda_t|)^2}.$$

This in turn gives a condition on $\lambda_t$ since the numerator needs to be strictly greater than 0 for this to make sense:

$$\frac{\phi(1 - K_1 \lambda_t)}{L} - \frac{2K_2|\lambda_t|}{\mu} > 0.$$

We see that this new condition is consistent with the past condition that $(1 - \lambda_t K_1) > 0$. Finally, we can write down the rate of linear convergence:

$$\|x_{t+1} - x^*\|_2^2 \leq (1 - (2 - \phi)\mu(1 - K_1 \lambda_t)\eta_t) \|x_t - x^*\|_2^2$$
$$= \left[ 1 - \frac{(2 - \phi)\mu(1 - K_1 \lambda_t)\left[ \frac{\phi}{L}(1 - K_1 \lambda_t) - \frac{2K_2|\lambda_t|}{\mu} \right]}{(1 - K_1 \lambda_t + K_2|\lambda_t|)^2} \right] \|x_t - x^*\|_2^2.$$

Similar to the proof of Theorem 14 in Appendix B.4, this is a linear rate of convergence due to $\epsilon$ and this rate is at best as good as gradient descent since $K_2|\lambda_t| \geq 0$. $\qquad \square$

## C  Missing Proofs in Section 5 Smooth and Convex Optimization

### C.1  Proof of Theorem 17

*Proof.* By similar reasoning as Corollary 15, we can reduce to the single dimensional case. We start by applying $L$-smoothness.

$$f(x_{t+1}) - f(x_t) \leq f'(x_t)(x_{t+1} - x_t) + \frac{L}{2}(x_{t+1} - x_t)^2$$
$$= -\eta f'(x_t)^C\delta_{c_t}^{\alpha,\beta}f(x_t) + \frac{L\eta^2}{2}(^C\delta_{c_t}^{\alpha,\beta}f(x_t))^2.$$

We bound the first term as:

$$-\eta f'(x_t)^C\delta_{c_t}^{\alpha,\beta}f(x_t) \leq -\eta f'(x_t)^2 + \eta\lambda K_1 f'(x_t)^2 + \eta|\lambda|K_2 f'(x_t)^2$$
$$= -\eta(1 - \lambda K_1 - |\lambda|K_2)f'(x_t)^2.$$

We bound the second term as:

$$(^C\delta_{c_t}^{\alpha,\beta}f(x_t))^2 \leq (|1 - \lambda K_1| + |\lambda|K_2)^2 f'(x_t)^2.$$

Putting everything together yields:

$$f(x_{t+1}) - f(x_t) \leq (-\eta(1 - \lambda K_1 - |\lambda|K_2) + \frac{L\eta^2}{2}(|1 - \lambda K_1| + |\lambda|K_2)^2)f'(x_t)^2.$$

For this to converge, we need the RHS to be negative which means that $1 - \lambda K_1 - |\lambda|K_2 > 0$. Therefore, $1 - \lambda K_1 > |\lambda|K_2 > 0$. Now, we use 3 point identity to proceed. Note this is where the case $p \neq 1$ breaks down since the $L^{1+p}$ norm is not induced by an inner product if $p \neq 1$.

$$(x_{t+1} - x^*)^2 = (x_{t+1} - x_t)^2 + 2(x_{t+1} - x_t)(x_t - x^*) + (x_t - x^*)^2$$
$$= \eta^2(^C\delta_{c_t}^{\alpha,\beta}f(x_t))^2 - 2\eta^C\delta_{c_t}^{\alpha,\beta}f(x_t)(x_t - x^*) + (x_t - x^*)^2.$$

We now bound the middle term on the RHS. Note that the following only works due to $f$ being convex and single dimensional.

$$-2\eta^C\delta_{c_t}^{\alpha,\beta}f(x_t)(x_t - x^*) \leq 2\eta|\lambda|K_2|f'(x_t)||x_t - x^*| - 2\eta f'(x_t)(1 - \lambda K_1)(x_t - x^*)$$
$$= 2\eta|\lambda|K_2(f'(x_t))(x_t - x^*) - 2\eta f'(x_t)(1 - \lambda K_1)(x_t - x^*)$$
$$= -2\eta(1 - \lambda K_1 - |\lambda|K_2)f'(x_t)(x_t - x^*)$$
$$\leq -2\eta(1 - \lambda K_1 - |\lambda|K_2)(f(x_t) - f(x^*)).$$

Putting everything together gives a bound on $f(x_t) - f(x^*)$.

$$f(x_t) - f(x^*) \leq \frac{1}{2\eta(1 - \lambda K_1 - |\lambda|K_2)}((x_t - x^*)^2 - (x_{t+1} - x^*)^2) + \frac{\eta(1 - \lambda K_1 + |\lambda|K_2)^2}{2(1 - \lambda K_1 - |\lambda|K_2)}(f'(x_t))^2.$$

Combining this with the bound on $f(x_{t+1}) - f(x_t)$ yields:

$$f(x_{t+1}) - f(x^*) \leq \frac{1}{2\eta(1 - \lambda K_1 - |\lambda|K_2)}((x_t - x^*)^2 - (x_{t+1} - x^*)^2)$$
$$+ \eta\left[\frac{(1 - \lambda K_1 + |\lambda|K_2)^2}{2(1 - \lambda K_1 - |\lambda|K_2)} - (1 - \lambda K_1 - |\lambda|K_2) + \frac{L\eta}{2}(1 - \lambda K_1 + |\lambda|K_2)^2\right](f'(x_t))^2.$$

Now, we derive the learning rate $\eta$ as follows so that the $f'(x_t)^2$ terms vanish.

$$\frac{L\eta}{2}(1 - \lambda K_1 + |\lambda|K_2)^2 = (1 - \lambda K_1 - |\lambda|K_2) - \frac{(1 - \lambda K_1 + |\lambda|K_2)^2}{2(1 - \lambda K_1 - |\lambda|K_2)}$$

$$\implies \eta = \frac{1}{L}\left[\frac{2(1 - \lambda K_1 - |\lambda|K_2)}{(1 - \lambda K_1 + |\lambda|K_2)^2} - \frac{1}{1 - \lambda K_1 - |\lambda|K_2}\right].$$

We require this learning rate to be positive (since this was assumed throughout the proof) so we get a condition on $\lambda$.

$$\left[\frac{2(1 - \lambda K_1 - |\lambda|K_2)}{1 - \lambda K_1 + |\lambda|K_2)^2} - \frac{1}{1 - \lambda K_1 - |\lambda|K_2}\right] > 0$$

$$\implies \left(\frac{1 - \lambda K_1 - |\lambda|K_2}{1 - \lambda K_1 + |\lambda|K_2}\right)^2 > \frac{1}{2}$$

$$\implies (1 - \lambda K_1)(1 - \frac{1}{\sqrt{2}}) - |\lambda|K_2(1 + \frac{1}{\sqrt{2}}) > 0$$

$$\implies 1 - \lambda K_1 > |\lambda|K_2\frac{\sqrt{2} + 1}{\sqrt{2} - 1}.$$

We can now write the earlier bound as:

$$f(x_{t+1}) - f(x^*) \le \frac{L}{\left(4\left(\frac{1 - \lambda K_1 - |\lambda|K_2}{1 - \lambda K_1 + |\lambda|K_2}\right)^2 - 2\right)}((x_t - x^*)^2 - (x_{t+1} - x^*)^2).$$

Applying telescope sum and bounding the LHS using convexity gives the desired result:

$$f(\bar{x_T}) - f(x^*) \le \frac{L(x_0 - x^*)^2}{\left(4\left(\frac{1 - \lambda K_1 - |\lambda|K_2}{1 - \lambda K_1 + |\lambda|K_2}\right)^2 - 2\right)T}.$$

This rate is at best the same as standard gradient descent since $|\lambda|K_2 \ge 0$. $\qquad\square$

## C.2  Proof of Theorem 18

*Proof.* We begin with $L$-smoothness.

$$f(x_{t+1}) - f(x_t) \le \langle \nabla f(x_t), x_{t+1} - x_t \rangle + \frac{L}{2}\|x_{t+1} - x_t\|_2^2$$

$$= -\eta_t\langle \nabla f(x_t), {}^C\delta_{c_t}^{\alpha,\beta}f(x_t)\rangle + \frac{L\eta_t^2}{2}\|{}^C\delta_{c_t}^{\alpha,\beta}f(x_t)\|_2^2.$$

We bound the first term as:

$$-\eta_t\langle \nabla f(x_t), {}^C\delta_{c_t}^{\alpha,\beta}f(x_t)\rangle \le -\eta_t(1 - \lambda_t K_1 - |\lambda_t|K_2)\|\nabla f(x_t)\|_2^2.$$

We bound the second term as:

$$\|{}^C\delta_{c_t}^{\alpha,\beta}f(x_t)\|_2^2 \le (|1 - \lambda_t K_1| + |\lambda_t|K_2)^2\|\nabla f(x_t)\|_2^2.$$

Putting everything together yields:

$$f(x_{t+1}) - f(x_t) \le (-\eta_t(1 - \lambda_t K_1 - |\lambda_t|K_2) + \frac{L\eta_t^2}{2}(|1 - \lambda_t K_1| + |\lambda_t|K_2)^2)\|\nabla f(x_t)\|_2^2.$$

For this to converge, we need the RHS to be negative which means that $1 - \lambda_t K_1 > |\lambda_t|K_2 > 0$. Now, we use 3 point identity to proceed.

$$\|x_{t+1} - x^*\|_2^2 = \|x_{t+1} - x_t\|_2^2 + 2\langle x_{t+1} - x_t, x_t - x^*\rangle + \|x_t - x^*\|_2^2$$

$$= \eta_t^2\|{}^C\delta_{c_t}^{\alpha,\beta}f(x_t)\|_2^2 - 2\eta_t\langle {}^C\delta_{c_t}^{\alpha,\beta}f(x_t), x_t - x^*\rangle + \|x_t - x^*\|_2^2.$$

For bounding the second term (note $|\cdot|$ is taken element-wise, $\eta_t \geq 0$ is assumed):

$$
\begin{aligned}
-2\eta_t \langle {}^C\delta_{c_t}^{\alpha,\beta} f(x_t), (x_t - x^*)\rangle &\leq 2\eta_t K_2 |\lambda_t| \langle |\nabla f(x_t)|, |x_t - x^*|\rangle - 2\eta_t(1 - \lambda_t K_1)\langle \nabla f(x_t), x_t - x^*\rangle \\
&\leq 2\eta_t K_2 |\lambda_t| \|\nabla f(x_t)\|_2 \|x_t - x^*\|_2 - 2\eta_t(1 - \lambda_t K_1)\langle \nabla f(x_t), x_t - x^*\rangle \\
&\leq 2\eta_t K_2 |\lambda_t| L \|x_t - x^*\|_2^2 - 2\eta_t(1 - \lambda_t K_1)(f(x_t) - f(x^*)).
\end{aligned}
$$

Putting everything together gives a bound on $f(x_t) - f(x^*)$.

$$
\begin{aligned}
f(x_t) - f(x^*) \leq{}& \frac{1}{2\eta_t(1 - \lambda_t K_1)}(\|x_t - x^*\|_2^2 - \|x_{t+1} - x^*\|_2^2) + \frac{|\lambda_t| K_2 L}{1 - \lambda_t K_1}\|x_t - x^*\|_2^2 \\
&+ \frac{\eta_t(1 - \lambda_t K_1 + |\lambda_t| K_2)^2}{2(1 - \lambda_t K_1)}\|\nabla f(x_t)\|_2^2.
\end{aligned}
$$

Combining this with the bound on $f(x_{t+1}) - f(x_t)$ yields:

$$
\begin{aligned}
f(x_{t+1}) - f(x^*) \leq{}& \frac{1}{2\eta_t(1 - \lambda_t K_1)}(\|x_t - x^*\|_2^2 - \|x_{t+1} - x^*\|_2^2) + \frac{|\lambda_t| K_2 L}{1 - \lambda_t K_1}\|x_t - x^*\|_2^2 \\
&+ \eta_t \Bigg[\frac{(1 - \lambda_t K_1 + |\lambda_t| K_2)^2}{2(1 - \lambda_t K_1)} - (1 - \lambda_t K_1 - |\lambda_t| K_2) \\
&+ \frac{L\eta_t}{2}(1 - \lambda_t K_1 + |\lambda_t| K_2)^2 \Bigg]\|\nabla f(x_t)\|_2^2.
\end{aligned}
$$

Solving for $\eta_t$ such that the $\|\nabla f(x_t)\|_2^2$ terms vanish yields:

$$
\eta_t = \frac{1}{L}\left[\frac{2(1 - \lambda_t K_1 - |\lambda_t| K_2)}{(1 - \lambda_t K_1 + |\lambda_t| K_2)^2} - \frac{1}{1 - \lambda_t K_1}\right].
$$

The proof thus far assumes that $\eta_t > 0$ which creates a constraint on $\lambda_t$.

$$
\begin{aligned}
&\left[\frac{2(1 - \lambda_t K_1 - |\lambda_t| K_2)}{(1 - \lambda_t K_1 + |\lambda_t| K_2)^2} - \frac{1}{1 - \lambda_t K_1}\right] > 0 \\
\implies{}& \frac{(1 - \lambda_t K_1 - |\lambda_t| K_2)(1 - \lambda_t K_1)}{(1 - \lambda_t K_1 + |\lambda_t| K_2)^2} > \frac{1}{2} \\
\implies{}& (1 - \lambda_t K_1)^2 - 4|\lambda_t| K_2(1 - \lambda_t K_1) - |\lambda_t|^2 K_2^2 > 0 \\
\implies{}& (1 - \lambda_t K_1 - 2|\lambda_t| K_2)^2 > 5|\lambda_t|^2 K_2^2 \\
\implies{}& 1 - \lambda_t K_1 > (\sqrt{5} + 2)|\lambda_t| K_2.
\end{aligned}
$$

Now, suppose $1 - \lambda_t K_1 = s_t |\lambda_t| K_2$ ($s_t > \sqrt{5} + 2$). Then the following useful equation holds:

$$
\begin{aligned}
&\eta_t = \frac{1}{L}\left[\frac{2(s_t - 1)}{(s_t + 1)^2 |\lambda_t| K_2} - \frac{1}{s_t |\lambda_t| K_2}\right] \\
\implies{}& \eta_t = \frac{1}{L s_t |\lambda_t| K_2}\left[\frac{s_t^2 - 4s_t - 1}{(s_t + 1)^2}\right] \\
\implies{}& s_t \eta_t |\lambda_t| K_2 = \frac{1}{L}\left[\frac{s_t^2 - 4s_t - 1}{(s_t + 1)^2}\right].
\end{aligned}
$$

Returning to the bound on $f(x_{t+1}) - f(x^*)$, with the chosen $\eta_t$, we have:

$$
f(x_{t+1}) - f(x^*) \leq \left(\frac{1}{2\eta_t(1 - \lambda_t K_1)} + \frac{|\lambda_t| K_2 L}{1 - \lambda_t K_1}\right)\|x_t - x^*\|_2^2 - \frac{1}{2\eta_t(1 - \lambda_t K_1)}\|x_{t+1} - x^*\|_2^2.
$$

For this sum to telescope, we need the following condition to hold:

$$\frac{1}{2\eta_{t+1}(1 - \lambda_{t+1}K_1)} + \frac{|\lambda_{t+1}|K_2 L}{1 - \lambda_{t+1}K_1} \leq \frac{1}{2\eta_t(1 - \lambda_t K_1)}$$

$$\implies \frac{1}{2\eta_{t+1}s_{t+1}|\lambda_{t+1}|K_2} + \frac{|\lambda_{t+1}|K_2 L}{s_{t+1}|\lambda_{t+1}|K_2} \leq \frac{1}{2\eta_t(s_t|\lambda_t|K_2)}$$

$$\implies \frac{(s_{t+1} + 1)^2}{s_{t+1}^2 - 4s_{t+1} - 1} + \frac{2}{s_{t+1}} \leq \frac{(s_t + 1)^2}{s_t^2 - 4s_t - 1}.$$

For $s_t > \sqrt{5} + 2$, both the LHS and RHS are decreasing functions that going from $\infty \to 1$ as $s_t \to \infty$. Thus if $s_{t+1}$ is chosen large enough, it will satisfy the telescoping condition. To solve for an exact cutoff would require solving a cubic equation so for practical use it is better to use a numerical solver. Now, assuming this condition holds, applying telescope sum to the $f(x_{t+1}) - f(x^*)$ inequality yields the desired result:

$$f(\bar{x_T}) - f(x^*) \leq \left( \frac{1}{2\eta_0(1 - \lambda_0 K_1)} + \frac{|\lambda_0|K_2 L}{1 - \lambda_0 K_1} \right) \frac{\|x_0 - x^*\|_2^2}{T}$$

$$= \frac{L}{2} \left[ \frac{(s_0 + 1)^2}{s_0^2 - 4s_0 - 1} + \frac{2}{s_0} \right] \frac{\|x_0 - x^*\|_2^2}{T}.$$

$\square$

# D    Missing Proofs in Section 6 Smooth and Non-Convex Optimization

## D.1    Proof of Theorem 20

*Proof.* Throughout the proof we will use the notation $[a_i]$ to denote a vector of $k$ elements with $i$th element $a_i$. We note that all the results for single variable $f$ hold since $f$ satisfies the single variable $(L, p)$-Hölder smooth definition in each component. We start with the $(L, p)$-Hölder smooth property:

$$f(x_{t+1}) - f(x_t) \leq \langle \nabla f(x_t), x_{t+1} - x_t \rangle + \frac{L}{1 + p}\|x_{t+1} - x_t\|_{p+1}^{p+1}$$

$$= -\eta \langle \nabla f(x_t), {}_p^C \delta_{c_t}^\alpha f(x_t) \rangle + \frac{L\eta^{p+1}}{1 + p}\|{}_p^C \delta_{c_t}^\alpha f(x_t)\|_{p+1}^{p+1}$$

$$\leq -\eta \left\langle \left[ \frac{\partial f}{\partial x^{(i)}}(x_t) \right], \left[ \frac{\partial f}{\partial x^{(i)}}(x_t)|x_t^{(i)} - c_t^{(i)}|^{1-p} \right] \right\rangle$$

$$\quad + \eta \left\langle \left[ \left| \frac{\partial f}{\partial x^{(i)}}(x_t) \right| \right], K|x_t^{(i)} - c_t^{(i)}| \right\rangle + \frac{L\eta^{p+1}}{1 + p}\|{}_p^C \delta_{c_t}^\alpha f(x_t)\|_{p+1}^{p+1}$$

$$= -\eta(\lambda^{1-p} - K\lambda) \sum_{i=1}^k \left| \frac{\partial f}{\partial x^{(i)}}(x_t) \right|^{1+1/p} + \frac{L\eta^{p+1}}{1 + p}\|{}_p^C \delta_{c_t}^\alpha f(x_t)\|_{p+1}^{p+1}$$

$$= -\eta(\lambda^{1-p} - K\lambda)\|\nabla f(x_t)\|_{1+1/p}^{1+1/p} + \frac{L\eta^{p+1}}{1 + p}\|{}_p^C \delta_{c_t}^\alpha f(x_t)\|_{p+1}^{p+1}$$

$$\leq -\eta(\lambda^{1-p} - K\lambda)\|\nabla f(x_t)\|_{1+1/p}^{1+1/p}$$

$$\quad + \frac{L\eta^{p+1}}{1 + p}\left\| \left[ \left| \frac{\partial f}{\partial x^{(i)}}(x_t)|x_t^{(i)} - c_t^{(i)}|^{1-p} \right| + K|x_t^{(i)} - c_t^{(i)}| \right] \right\|_{p+1}^{p+1}$$

$$\leq -\eta(\lambda^{1-p} - K\lambda)\|\nabla f(x_t)\|_{1+1/p}^{1+1/p}$$

$$\quad + \frac{L\eta^{p+1}}{1 + p}(\lambda^{1-p} + K\lambda)^{p+1}\left\| \left[ \left| \frac{\partial f}{\partial x^{(i)}}(x_t) \right|^{1/p} \right] \right\|_{p+1}^{p+1}$$

$$= -\eta(\lambda^{1-p} - K\lambda)\|\nabla f(x_t)\|_{1+1/p}^{1+1/p} + \frac{L\eta^{p+1}}{1 + p}(\lambda^{1-p} + K\lambda)^{p+1}\|\nabla f(x_t)\|_{1+1/p}^{1+1/p}$$

$$= -\eta(\lambda^{1-p} - K\lambda)\|\nabla f(x_t)\|_{1+1/p}^{1+1/p} + \frac{L\eta^{p+1}}{1+p}(\lambda^{1-p} + K\lambda)^{p+1}\|\nabla f(x_t)\|_{1+1/p}^{1+1/p}$$

$$= -\psi\|\nabla f(x_t)\|_{1+1/p}^{1+1/p}.$$

For this to converge, we need $\psi > 0$ which gives a condition on $\eta$ as follows:

$$\lambda^{1-p} - K\lambda - \frac{L}{1+p}\eta^p(\lambda^{1-p} + K\lambda)^{1+p} > 0$$

$$\implies \eta^p < \frac{(1+p)(\lambda^{1-p} - K\lambda)}{L(\lambda^{1-p} + K\lambda)^{1+p}}$$

$$\implies \eta < \sqrt[p]{\frac{(1+p)(\lambda^{1-p} - K\lambda)}{L(\lambda^{1-p} + K\lambda)^{1+p}}}.$$

This in turn gives a condition on $\lambda$ in order to make the interior of the root positive:

$$(\lambda^{1-p} - K\lambda) > 0$$

$$\implies \lambda^p < \frac{1}{K}$$

$$\implies \lambda < \sqrt[p]{\frac{1}{K}}.$$

We derive the convergence bound as follows:

$$\min_{0 \le t \le T}\|\nabla f(x_t)\|_{1+1/p}^{1+1/p} \le \frac{1}{T+1}\sum_{t=0}^{T}\|\nabla f(x_t)\|_{1+1/p}^{1+1/p}$$

$$\le \sum_{t=0}^{T}\frac{f(x_t) - f(x_{t+1})}{(T+1)\psi}$$

$$= \frac{f(x_0) - f(x_{T+1})}{(T+1)\psi}$$

$$\le \frac{f(x_0) - f^*}{(T+1)\psi}.$$

$\square$

# E  Missing Proofs in Section 7 Finding the Advantage of Fractional Gradient Descent

## E.1  Proof of Theorem 21

*Proof.* We first note some useful equations:

1. $\nabla f(x) = \mathbf{A}x + b$

2. ${}^C\delta_c^{\alpha,\beta}x = 1$ (for 1 dimensional x)

3. ${}^C\delta_c^{\alpha,\beta}b^T x = b$

4. ${}^C\delta_c^{\alpha,\beta}\frac{a}{2}x^2 = a(\beta - \gamma)(x - c) + ax$ (for 1 dimensional x)

5. ${}^C\delta_c^{\alpha,\beta}1 = 0$

Equation 1 is simple differentiation. Equation 2 follows since the 2nd derivative of $x$ is 0. Equation 3 follows by linearity of the fractional gradient descent operator and equation 2. Equation 4 follows directly from Lemma A.1 of Shin et al. (2021). Equation 5 follows since derivatives of a constant function are 0.

We now leverage these equations to apply the fractional gradient descent operator to $f$. These equations tell us that the only terms where the fractional gradient descent operator does not act like a normal gradient are the squares corresponding to diagonal elements of $\mathbf{A}$, $a_{ii}$. These elements look precisely like equation 4 and can be handled as follows:

$$
\begin{aligned}
{}^C\delta_{c^{(i)}}^{\alpha,\beta} \frac{a_{ii}}{2}\left(x^{(i)}\right)^2 &= a_{ii}(\beta - \gamma)(x^{(i)} - c^{(i)}) + a_{ii}x^{(i)} \\
&= a_{ii}(\beta - \gamma)\left(-\frac{\Delta}{(\beta - \gamma)L}(\nabla f(x))^{(i)}\right) + a_{ii}x^{(i)} \\
&= -a_{ii}\left(\frac{\Delta}{L}(\mathbf{A}x + b)^{(i)}\right) + \frac{d}{dx^{(i)}}\left(\frac{a_{ii}}{2}\left(x^{(i)}\right)^2\right).
\end{aligned}
$$

Putting everything together with linearity we calculate the full fractional gradient descent operator as follows:

$$
\begin{aligned}
{}^C\delta_c^{\alpha,\beta} f(x) &= \operatorname{diag}([-a_{11}\frac{\Delta}{L}, ..., -a_{kk}\frac{\Delta}{L}])(\mathbf{A}x + b) + \nabla f(x) \\
&= \operatorname{diag}([-a_{11}\frac{\Delta}{L}, ..., -a_{kk}\frac{\Delta}{L}])(\mathbf{A}x + b) + (\mathbf{A}x + b) \\
&= (\mathbf{DA})x + \mathbf{D}b \\
&= \mathbf{A}'x + b'.
\end{aligned}
$$

Thus we have the desired result for point 1 of the theorem. For point 2, we start by noting that ${}^C\delta_c^{\alpha,\beta} f(x^*) = 0$ when $\mathbf{DA}x^* = \mathbf{D}b$. Since $\mathbf{A}$ and $\mathbf{D}$ are both invertible, we see that $x^* = \mathbf{A}^{-1}b$. At this point, we also have $\nabla f(x^*) = 0$. This tells us that if fractional gradient descent converges, it converges to the right point.

The update rule for fractional gradient descent can be written as $x_{t+1} = x_t - \eta(\mathbf{A}'x_t + b')$. For proving convergence, we begin with the 3 point equality:

$$
\begin{aligned}
\|x_{t+1} - x^*\|_2^2 &= \|x_{t+1} - x_t\|_2^2 + 2\langle x_{t+1} - x_t, x_t - x^*\rangle + \|x_t - x^*\|_2^2 \\
&= \eta^2\|\mathbf{A}'x_t + b'\|_2^2 - 2\eta\langle\mathbf{A}'x_t + b', x_t - x^*\rangle + \|x_t - x^*\|_2^2 \\
&= \eta^2\|\mathbf{A}'x_t + b'\|_2^2 - 2\eta\langle\mathbf{A}'x_t + b' - \mathbf{A}'x^* - b', x_t - x^*\rangle + \|x_t - x^*\|_2^2 \\
&= \eta^2\|\mathbf{A}'x_t + b'\|_2^2 - 2\eta(x_t - x^*)^T\mathbf{A}'(x_t - x^*) + \|x_t - x^*\|_2^2 \\
&\le \eta^2\|\mathbf{A}'x_t + b'\|_2^2 - \eta\mu'\|x_t - x^*\|_2^2 - \eta(x_t - x^*)^T(\mathbf{A}'x_t + b' - \mathbf{A}'x^* - b') + \|x_t - x^*\|_2^2 \\
&= \eta^2\|\mathbf{A}'x_t + b'\|_2^2 - \eta\mu'\|x_t - x^*\|_2^2 - \eta(\mathbf{A}x_t + b' - \mathbf{A}x^* - b')^T\mathbf{A}'^{-1\mathbf{T}}(\mathbf{A}'x_t + b') + \|x_t - x^*\|_2^2 \\
&= \eta^2\|\mathbf{A}'x_t + b'\|_2^2 - \eta\mu'\|x_t - x^*\|_2^2 - \eta(\mathbf{A}x_t + b')^T\mathbf{A}'^{-1\mathbf{T}}(\mathbf{A}'x_t + b') + \|x_t - x^*\|_2^2 \\
&\le \eta^2\|\mathbf{A}'x_t + b'\|_2^2 - \eta\mu'\|x_t - x^*\|_2^2 - \frac{\eta}{L'}\|\mathbf{A}x_t + b'\|_2^2 + \|x_t - x^*\|_2^2 \\
&= \left(\eta^2 - \frac{\eta}{L'}\right)\|\mathbf{A}'x_t + b'\|_2^2 + (1 - \eta\mu')\|x_t - x^*\|_2^2.
\end{aligned}
$$

Setting $\eta = \frac{1}{L'}$ similarly to in standard gradient descent, we arrive at the desired convergence rate:

$$
\|x_{t+1} - x^*\|_2^2 \le \left[1 - \frac{\mu'}{L'}\right]\|x_t - x^*\|_2^2.
$$

$\square$

