# OpenReview forum: "Convergence Analysis of Fractional Gradient Descent"
_TMLR — Accepted by TMLR_

### Review · Reviewer_4UxH · 2023-12-15

**Summary Of Contributions:**

The paper extends the study of fractional derivatives for optimization. Some approximaition results are derived connecting first order derivatives with the fractional Caputo derivatives. The results are then applied to derive some guarantees for convex, strongly convex and non-convex settings. Preliminary empirical results are presented comparing convergence and proposed learning rates to those of standard GD.

**Audience:**

Yes

**Broader Impact Concerns:**

-

**Claims And Evidence:**

Yes

**Requested Changes:**

See weaknesses section above.

**Strengths And Weaknesses:**

**Strengths**:

The paper studies mathematically interesting question of understanding convergence of fractional gradient methods, and extends the convergence guarantees to standard optimization settings, e.g., convex, non-convex.

**Weaknesses (major)**:
1. The proposed algorithm, e.g., in Theorem 10 is only conceptual. It is unclear how to select the sequence c_t. Moreover, I find it weird that the proposed step-size sequence can become negative or infinity (when $c_t = x_t$).

**Weaknesses (minor)**:
1. On page 3, the objects like $g$ and $\psi$ suddenly appear and are not properly defined. Moreover, it would make more sense to defer this inequality to the proof of Corollary 5.
2. It should be properly discussed how Caputo derivative can be computed in practice (e.g., using Gauss-Jacobi quadrature) and if it coincides  with the integer order derivatives, e.g., for $\alpha = 0, 1, 2$. This is not obvious from Definition 3.
3. Definition 6 and 7 have been studied before in the context of usual gradient descent and have well-established names in the literature. Def. 6 is called Hölder smoothness, see e.g., Nesterov (2015).
4. In Section 3, the approximation result (of Caputo derivative with the usual first order derivative) is presented for Hölder smooth and uniformly convex functions, but starting from the next section this generality is dropped and standard smoothness and strong convexity is considered. Could the authors comment what are the challenges for handling a more general setting?
5. The paper would benefit from a separate “methods” section devoted to describing the proposed algorithms with specific choice of the sequences c_t and the stepsizes.
6. It is unclear why the proposed algorithms for convex, strongly convex and non-convex settings differ so much.
7. What is the difference between AT-CFGD in Shin et al (2021)? What is the intuition for why Fractional Descent is faster on a quadratic. Can it be simply attributed to the fact that this algorithm has more parameters to tune? Intuitively when decreasing $\alpha$, the rate should become worse.


Nesterov, Y. Universal gradient methods for convex optimization problems. Math. Program. 152, 381–404 (2015).


**Questions**:

1. In Figure 1, the optimal step-size refers to the optimal according to theory or the one found by fine-tuning?

2. In Section 5.1, some results are generalized from 1 dimensional to “high dimensional” case, i.e., k > 1. However, this is done in very limiting separable case. Could the authors elaborate more on the challenges for a more general setting?

Typo: Page 1 “integral order derivatives” -> “integer order derivatives”

---

> ### Author Response · Authors · 2024-01-16
> **Response**
>
> Dear Reviewer,
>
> Thank you for your detailed comments and your time providing this review.  I have taken actions to address all of the listed weaknesses as follows.
>
> Weaknesses (Major):
> 1) I agree that the result of Theorem 10 was rather weak.  I have replaced it with Theorem 17 and 18 that examine fractional gradient descent on smooth and convex functions in much more detail including specifically how to select $c_t$.  The reason the step size had that odd behavior was because as seen in the new Theorem 4, there is a sign correction needed to compare the fractional derivative with f'(x).  The behavior as $c_t\to x_t$ of the step size exploding was to compensate for the fractional derivative vanishing.  Now, these issues are less relevant since the fractional gradient descent method has a coefficient dependent on $x_t-c_t$.
>
> Weaknesses (Minor):
> 1) I agree that discussion was distracting and I have removed it
> 2) I have added a note with a reference (Shin et al (2021)) that goes through the details of Gauss-Jacobi quadrature for computing fractional gradient descent.  The newly added Theorem 4 covers the behavior of the Caputo derivative as the order of the derivative approaches integers.
> 3) Thank you for pointing this out, I have integrated the proper terminology wherever it was used.
> 4) The only setting in which Hölder smoothness is used is for smooth and non-convex functions.  The issue with the other two settings - smooth and convex, and smooth and strongly convex - is that when trying to generalize to Hölder smooth and uniformly convex functions, the proof breaks due to $L^{1+p}$ norms not being induced by an inner product unless $p=1$.  Roughly speaking, a crucial step is of the form $|a-b|^2 = |a-c|^2+2<a-c,c-b>+|c-b|^2$.  The middle inner product is usually negative which allows for convergence results to be proven.  I have been unable to come up with a suitable replacement for this equation when applying it to $|a-b|^{1+p}$ and as such, I leave the generalization for these two settings to future work.
> 5) I agree that clarity about the methods needs improvement so I have added methods subsections and made it so there are only two methods (the 2nd being a variation of the 1st needed for the proof of the Hölder smooth and convex setting).  Every convergence theorem necessarily employs different step sizes which are derived in the proofs of the theorems.  As such, I have left the specific formulas for parameters to the statement of the theorems.
> 6) I agree that the variations and reasons for the variations were unclear.  As such, the strongly convex and convex settings now share a single method.  The nonconvex setting with Hölder smoothness necessarily needs to restrict and extend this method slightly which I have explained in the revised paper.
> 7) The main difference between the strongly convex and convex settings' method and AT-CFGD in Shin et al (2021) is in how the terminal point is chosen - I choose it based off of the gradient and AT-CFGD chooses it as a previous $x_t$ - which I have explained in the revised paper.  For now, it is difficult to do better than the more tunable parameters explanation since even for the same $L, \mu$ parameters, empirically, fractional gradient descent might be slower or faster than standard gradient descent depending on the function itself.  I think it is important in future work to nail down when fractional gradient descent can be faster than standard gradient descent using additional assumptions on the function.
>
> Questions
> 1) The optimal step size refers not to theoretical step size, but instead to the optimal step size derived by considering the optimal step in the direction given by fractional gradient descent (which has a closed form solution for quadratic functions from Shin et al (2021)).
> 2) In section 5.1 (now section 4.3 in the revision), there is a theorem (Theorem 16 in the revision) after the result for the separable case that gives a convergence analysis of the general case.  The challenge in this setting is one of the terms does not generalize well which causes the convergence analysis to become more complicated.  For more information, please take a look at Theorem 16 in the revision and the preceding discussion.
>
> Thank you for catching the typo, I have fixed it.
>
> Once more, thank you for your time in reviewing the paper and your helpful comments.

---

### Review · Reviewer_x1vw · 2024-01-25

**Summary Of Contributions:**

This paper studies the convergence of fractional gradient descent,
a modification of gradient descent that uses fractional derivatives
instead of the first-order "integer" gradient.
The analysis focuses on bounding the difference between fractional derivatives
and the first-order derivative for functions which are Hölder smooth and
uniformly convex (generalizations of Lipschitz-smooth and strongly convex).
Convergence of gradient descent is then used to deduce convergence of
the fractional gradient method, although at a slower rate due to these errors.
This approach allows the authors to prove convergence for uniformly convex,
convex, and non-convex functions.
Toy experiments on low-dimensional quadratics are presented to showcase
when fractional gradient descent outperforms gradient descent and vice-versa.

**Audience:**

Yes

**Broader Impact Concerns:**

None.

**Claims And Evidence:**

Yes

**Requested Changes:**

At a minimum, missing explanations/intuitions, such as for  $\beta$, should be
added to the text.
I don't want to push for additional theorems, such as lower bounds or
alternative analyses based on smoothness/convexity in terms of fractional
gradients, since these are out of scope for a short submission.
However, I do think some discussion of these issues could be raised in the
paper, particularly regarding the necessity of lower bounds for determining if
the theorems presented are tight.

**Strengths And Weaknesses:**

The main strength of this paper is the novelty of the fractional gradient scheme.
Although the convergence theorems presented have substantial drawbacks, this
may be mitigated by the fact that they are early contributions to a newly
developing sub-area in iterative optimization.
To summarize, the strengths and weaknesses are the following:

**Strengths**:

- The authors provide a novel convergence analysis analysis for fractional
    gradient descent on general convex and non-convex functions, going beyond
    previous work which focused on quadratic optimization.

- The analysis for fractional gradient descent in the non-convex setting
    makes novel use Hölder continuity and uses a simplified update.

- Experiments show fractional gradient descent can be significantly faster
    than gradient descent on toy problems.

**Weaknesses**:

- The convergence rates follow by bounding the difference between the fractional
    and first-order derivatives. This means that (a) the analysis doesn't
    give any intuition into fractional gradient descent and (b) the rates
    proved this way cannot improve upon the guarantees for gradient descent.

- The theoretical guarantees for scalar minimization (and separable minimization
    to a lesser degree) are not particularly interesting, given that bisection
    obtains a linear rate in this setting.

- Some algorithmic parameters, such as $\beta$, are not explained despite
    playing an integral role in the algorithm and analysis.

- The experiments are presented only for synthetic data and it is not clear why
    fractional gradient outperforms gradient descent when it does.


### Additional Details

The area of fractional gradient methods is new from an optimization
perspective, but the motivation for studying and using these methods
(beyond novelty) is not well developed.
The authors suggest fractional gradient descent should be investigated by
presenting an experiment (Figure 1) that demonstrates it can sometimes converge
faster than gradient descent.
However, they also later show that fractional gradient can vastly
under-perform gradient descent despite using a larger step-size (Figure 4).
Thus, the "better methods" argument is not quite sufficient for motivating
fractional gradient descent and I am left unsure of the motivation for
the paper.

A similar issue applies to the convergence results presented here.
The convergence rates for fractional gradient descent are uniformly worse
than those for gradient descent; while I applaud the authors for being
straightforward about this weakness, I am left with two worries:
(i) fractional gradient descent is a slow optimization method
or (ii) the wrong assumptions have been used in the theoretical analysis.
This second issue is particularly noticeable, as much of the analysis feels,
to use an analogy, "like a square peg being inserted into a round hole".
I think the authors also feel this tension, as they comment that one way to
improve their results is to "[use] convexity and smoothness definitions that
only involve fractional derivatives".
I think addressing one or both of these issues would greatly improve the paper.

Finally, I think the paper suffers from a lack of explanation and clarity.
For example, no intuition is given for the main fractional gradient update
on page 5. In particular, the parameter $\beta$ is not explained nor is the
necessity for a higher-order fractional derivative motivated.
Similarly, none of the convergence rates are interpreted or compared against
rates for standard methods like gradient descent.
Providing context of this kind is important for readability of the paper.

### Minor Issues:

Figure 1:
- I suggest using LaTeX for the label of the y-axis. This is supported natively
    by matplotlib.
- The font-size of the figure labels, ticks, etc. should be increased. In general,
    font-sizes for figures should be as at least as large as the font-size used
    in the main text.
- Rather than plot the logarithm of the objective along the y-axis, I suggest
    plotting the objective in log-scale. This will be more readable.

Page 2:
- Since the list of contribution is ordered, you might want to use "enumerate"
    rather than "itemize".

Definitions 1-3: It's a somewhat confusing to use the same notation for
    the left derivative, the right derivative, the unified Caputo derivative.

Appendix A.2: This is properly spelt L'Hopital's rule.

Definitions 7, 8: Nesterov a uses slightly different notion of Hölder continuity
    and uniform convexity that where the power of the norm is raised to $p$,
    but the norm itself is fixed. This allows Hölder continuity to interpolate
    between Lipschitz continuity of $f$ and of $\nabla f$ _in the same norm_.
    What is the importance of changing the norm in your definition?

Section 4.1:

- What is the purpose of developing Hölder continuity in full generality
    if only $p = 1$ is used? Then smoothness and strong convexity would be
    sufficient, right?

- Can you provide more insight into the definition of the fractional gradient
    method? What is $\beta$ and why do we include the $\alpha+1$ order
    fractional derivative in the descent step?

- Does the inclusion of the $\alpha + 1$ fractional derivative imply that
    the fractional gradient method is somewhere between a first and second
    order method?

- It appears from Theorems 12 and 13 that $\beta$ can be negative or positive
    or even zero. Since $\beta$ controls the role of the higher-order fractional
    gradient in the update, I would have thought $\beta > 0$ was strictly
    necessary.
    If this isn't true, how should $\beta$ be chosen?
    Looking at Theorem 16, is there a clear value of $\beta$ minimizes the
    oracle complexity?

Theorem 20:
- This result does not show convergence to a local optimum; it only shows
    convergence to a critical point where the gradient is $0$. This could
    be a saddle-point or even a local maximum.


Figure 3:
- I don't know what "Guided by Gradient" means.

- You should also plot the performance of the theoretical step-size for
    fractional gradient descent in the left-hand-side plot.


Section 8:
- It's not clear if the "existing data" from the assumptions is insufficient
    or if the theorems are simply weak. You need lower bounds to determine this
    fact.

- Defining smoothness and convexity in terms of fractional derivatives
    seems like the mos promising direction. This is the natural way to derive
    convergence rates in this setting.

---

> ### Author Response · Authors · 2024-01-31
> **Response**
>
> Dear Reviewer,
>
> Thank you for your review.  I will address each point below.
>
> Weaknesses:
>
> I agree that the assumptions and the way of bridging fractional derivatives and integer derivatives to use the assumptions make it difficult to prove advantage over gradient descent.  I think one of the main ways to overcome this is likely going to be through changing the assumptions to involve fractional derivatives directly.  I have added more details about this in the future directions section.  In addition, I have added Theorem 21 which can show a theoretical speed up over gradient descent in a limited quadratic setting.
>
> I agree that the single dimensional result is not particularly interesting.  Currently, I have kept it in since it serves as a good prototype for a lot of future results (especially in establishing a lot of useful tools in the proof), however, I am open to removing it if that is what you would recommend.
>
> I have added more details in Section 4.1 about the intuition behind $\beta$ in the fractional gradient descent method based on the paper which first introduced the method.  In principle, setting $\beta=0$ tends to either not change or help the convergence rates so one important future direction is to figure out if there is a better way to handle the $\beta$ term in the fractional gradient descent method (see the future directions section for more details).
>
> I have added Theorem 21 after the empirical results to analyze from a theoretical perspective why fractional gradient descent is able to do better than standard gradient descent.  While the results are only for quadratic functions and are somewhat limited, I think it provides useful insight into how fractional gradient descent can reduce the condition number and therefore improve convergence rates.
>
> Additional Details:
>
> I think Theorem 21 and the discussion surrounding it provides some insight into why the convergence is faster in some cases and the reason behind why Figure 4's convergence was significantly worse for fractional gradient descent.  Admittedly, proving fractional gradient descent has an advantage in more general cases has proven to be difficult and I will leave that for future work.
>
> I completely agree with the point about "like a square peg being inserted in a round hole".  I have addressed this point in the paper and added more details in the future directions about potential ways to overcome this such as considering Lipschitz continuity of the fractional gradient descent operator.
>
> I have tried to improve the readability by adding in explanations for a wide range of things ranging from $\beta$'s intuition to implementation details to future directions.  One of the main struggles I was having with readability previously was trying to keep everything within 12 pages which resulted in less detail that I would have liked.  In the current draft, I ended up going to 14.5 pages due to both additional results and explanations.  Please let me know if this is an issue and if so what parts I should try to cut out.  Thank you for your help.
>
> Minor Issues:
>
> Figure 1: I have re-generated all the figures with better font-size, latex, and log-scales when appropriate.
>
> Page 2:  Thank you for the suggestion I have switched to enumerate.
>
> Definition 1-3: I have added + and - superscripts for the left and right Caputo derivatives respectively.
>
> Appendix A.2:  Thank you for the catch, I have fixed the typo.
>
> Definitions 7,8: I have added a note in the paper pointing this out and explaining that it is easier to use the $1+p$ norm since having one term per dimension is very convenient for bounding things.
>
> Section 4.1: I use $p\neq 1$ in Section 6.  It is convenient to develop bounds for general $p$ since all cases can be handled together.  I added more details concerning the intuition behind $\beta$ based on the paper that introduced the method.  In general sending $\beta$ to zero tends to be better for the convergence rate.  There are two possibilities: one that the $\alpha+1$ derivative is not as useful given my choice of terminal point $c$ or that I need to use a better bound than the one in Lemma 11 when working with it.  I have discussed in the future directions section some possibilities as well as some difficulties I encountered when trying to better use the $\beta$ term.
>
> Theorem 20: Thank you for catching this.  It should have been convergence to a stationary point.  I have fixed it in the current revision.
>
> Figure 3: I agree this term was unclear.  I have defined it in Section 4.1 in the current revision to refer to the fractional gradient method.
>
> Section 8: I have updated the future directions to include lower bounds thank you for pointing that out.
>
> Requested Changes:
>
> I have added in explanations throughout and expanded the future directions sections significantly.  The page length increased from the limit of 12 to 14.5 during revision.  Please let me know if this is a problem.
>
> Once more, thank you for your time and review.

---

> > ### Comment · Reviewer_x1vw · 2024-02-09
> >
> > Thanks for your response and for updating the paper. I will read the modification sections shortly.
> >
> > -  Theorem 21 sounds like it addresses some of my worries about the paper. Even if it's a limited setting like quadratics, showing that fractional gradient descent can improve over first-order gradient descent is an important motivator for this work.
> >
> > - If choosing $\beta = 0$ doesn't really affect the convergence rate, what is the motivation for including the higher-order fractional derivative in the update? It seems both simpler in presentation and easier to compute the fractional gradient update when $\beta = 0$.

---

> > > ### Author Response · Authors · 2024-02-12
> > > **Reply**
> > >
> > > Dear Reviewer,
> > >
> > > Thank you for your comments.  Addressing the second point concerning $\beta$, the main reason for its inclusion stems from the prior work that originally defined the fractional gradient descent update.  The modification to that update I used in the paper was in choosing the terminal point of the fractional derivative, $c$, based on the gradient (see Section 4.1 for more details).  For motivation for $\beta$ in this modified update, if we take the limit of the fractional gradient descent update as $\alpha\to1$, we get that the update is proportional to $(1-\lambda\beta f''(x))f'(x)$ which is an approximation to $f'(x)/f''(x)$, a second order update, if $\beta\neq 0$.  It turns out for quadratic functions, we can eliminate the effects of changing $\alpha$ and $\beta$ entirely by choosing $\lambda$ properly as long as either $\alpha<1$ or $\beta \neq 0$.  What this means is that setting $\beta=0$ if $\alpha$ is non-integer is giving us a quasi-second order update without involving second derivatives.  One thing to note here is that due to how the fractional gradient descent update uses second derivatives, only the diagonal of the Hessian is ever accessed which means that the computation is somewhat simpler.  That said, this is also a limitation in that this approximate second order update makes less sense if the Hessian is not well approximated by a diagonal matrix and this intuition of being able to set $\beta=0$ is only shown thus far for quadratic functions.  For more general classes of functions, the strategy for proving rates in some sense treats the worst case in which $\beta$ slows down the optimization rate (doing better likely requires more assumptions than just $L$, $\mu$ since even quadratic functions require more assumptions for showing a speed up).  That said, I still think these worst case rates are useful since they show that on general functions, this fractional gradient descent method can still have guaranteed performance close to standard gradient descent.
> > >
> > > Please let me know if you have any further questions/comments or if there is anything I should revise.  Thank you.

---

### Review · Reviewer_6T9r · 2024-01-26

**Summary Of Contributions:**

The paper considers optimization by fractional gradient descent, where the Caputo derivative is used to construct the search direction. The paper first shows the quantitative connection between the first derivative and fractional derivative, and uses it to derive convergence rates of the fractional gradient descent for several problems, including smooth & convex problems, smooth & strongly convex problems and smooth and nonconvex problems. The paper also gives some experimental results to illustrate the behavior of fractional gradient descent versus gradient descent.

**Audience:**

Yes

**Broader Impact Concerns:**

I have no concerns on the ethical implications of the work.

**Claims And Evidence:**

No

**Requested Changes:**

I would like to see the discussions on the extensions to a larger $\alpha>1$, and show the advantage of the algorithm over gradient descent.

How to compute the step size efficiently in practical implementations?

The theoretical deductions should be explained more clearly.

**Strengths And Weaknesses:**

**Strength**

Fractional gradient descent is not well studied in the literature. The paper gives several convergence rates of fractional gradient descent. For smooth and convex problems, the paper derives convergence rates of order $O(1/T)$ after $T$ iterations. For smooth and strongly convex problems, the paper derives linear convergence.

The paper considers general strong convexity and general smoothness with $p\geq1$.


**Weakness**

- While the paper derives several convergence rates, these theoretical results of fractional gradient descent do not show the advantage of fractional gradient descent over gradient descent. Indeed, these convergence rates match those of the gradient descent.

- In Section 4 and Section 5, the paper considers the one-dimensional case. While Section 5.1 considers the high-dimensional case, it assumes the function to take a specific separable structure, i.e., f(x)=\sum_if_i(x^{(i)}), where $f_i$ is a one-dimensional function. This limits the application of theoretical analysis.

- The theoretical analysis focuses on $\alpha\in(0,1)$, which corresponds to a low-order derivative. This seems to be a bit restrictive. It would be interesting to extend the analysis to a larger $\alpha>1$, and show the advantage of the algorithm over gradient descent.

- The step size selection seems to be very complicated. For example, Theorem 14 requires very complicated assumptions on c_t and \eta_t. These assumptions make the implementation of the algorithm difficult.

- The theoretical analysis is not presented in a clear way. For example, in the proof of Theorem 10, I cannot follow clearly the way to bound $-\eta_tf'(x_t)^CD^\alpha_{c_t}f(x_t)$. It would be helpful if the authors can explain it more clearly.

- The experimental results seem to be not convincing. For example, I cannot see clearly the advantage of fractional gradient descent over gradient descent in Figure 4. The objective functions considered in the paper seem to be a bit simple.

---

> ### Author Response · Authors · 2024-01-31
> **Response**
>
> Dear Reviewer,
>
> Thank you for your review.  I will address each point below.
>
> Weaknesses:
>
> I agree that the lack of rates showing an advantage is a weakness of the paper.  Showing advantage with the standard smoothness/convexity settings proved to be very difficult.  I have, however, added some preliminary results on theoretical advantage for fractional gradient descent on quadratic functions in the latest update which can be found in Theorem 21.  Ultimately, I believe that using different assumptions is probably necessary to prove advantage in more general cases.
>
> There is exactly one term that makes going from separable to general functions difficult so I chose to split the two cases.  For the general higher dimensional case, please see Theorems 16 and 18 for smooth and strongly convex, and smooth and convex functions respectively.
>
> I agree that one of the lacking points of the analysis is the handling of the derivative for degree larger than 1.  In principle, the most interesting degree above 1 lies between 1 and 2 (since higher degrees intuitively do not directly optimize the function).  That said, even though the optimization method makes use of this derivative, it is difficult to show that it is effective for optimization.  I have devoted a part of the future directions section to discussing these difficulties and potential next steps.
>
> I agree, the formulas definitely do seem complicated at first glance.  I have added additional explanation to try to make implementation details more clear.  For most methods, we effectively have to do a similar approximation as in gradient descent of balancing keeping the learning rate low enough that the method converges and keeping it high enough that the convergence rate is still good.
>
> Theorem 10 in the original draft no longer exists.  It has been replaced by Theorems 17 and 18 that give more depth for the smooth and convex setting.  Please let me know there are any places in the proofs that are confusing and I will add more detail.
>
> Figure 4 is meant to serve as a counterexample that even with the same $L$, $\mu$ data, fractional gradient descent might do better or worse than gradient descent depending on more details of the actual objective function.  Theorem 21 and the following discussion give more details as to why this is happening and what the theory behind the speed-up given by fractional gradient descent on quadratic functions is.
>
> Requested Changes:
>
> As aforementioned, I have added a discussion about larger orders of derivative into the future work section as well as the challenges of using these derivatives effectively.  For showing advantage over gradient descent, I have added Theorem 21 which gives some basic results on quadratic functions.  One important direction of future study is to extend this result to more general functions and general methods.
>
> I have added implementation details throughout which should hopefully make this clearer.
>
> Please let me know if there are any other points of confusion in the theoretical deductions and I will add more details.
>
> Once again, thank you for your time and your review.
>
> As one last note, adding more explanations/results has caused the length to go from the standard 12 to 14.5.  Please let me know if this is a problem and if I should cut out some content to reduce the length.  Thank you.

---

> > ### Comment · Action_Editor_gwSb · 2024-02-10
> > **Reviewer, please engage with the authors**
> >
> > Dear Reviewer 6T9r,
> >
> > The authors have provided an extensive response to your feedback. Please comment as to whether it addresses your concerns or not.

---

> > ### Comment · Reviewer_6T9r · 2024-02-15
> > **Thank you for your reply**
> >
> > You added Theorem 21 to show an advantage of fractional gradient descent to gradient descent. There are several assumptions in the theorem. Can you give an explicit example of the matrix $A$ and $D$ such that all the assumptions hold?
> >
> > Also, the notation is a bit confusion. The notation $c$ is used in both the objective function and the computation of fractional gradient.

---

> > > ### Author Response · Authors · 2024-02-15
> > > **Reply**
> > >
> > > Dear Reviewer,
> > >
> > > Thank you for your comments.  Addressing the first point, for an example of $A$ where Theorem 21 can be applied please take a look at the discussion immediately following the theorem.  I examine the simplest case where $A$ is diagonal and show that $DA$ has a smaller condition number provided we choose $\Delta$ properly.  In this case, multiplication by $D$ corresponds to a function on the diagonal of $A$ of the form $a\to a(1-a\frac{\Delta}{L})$.
> > >
> > > I apologize for the notational confusions, using $c$ repeatedly was an oversight on my part.  I have updated the notation to use $y_0$ to reflect the value of the quadratic at the origin.  Please let me know if there is anything else that needs clarification.  Thank you.

---

### Decision · Action_Editor_gwSb · 2024-03-08

**Recommendation:** Accept with minor revision

**Comment:**

As a minor weakness, reviewers also pointed out that sometimes the paper is a bit hard to follow and that the experiments were performed on toy problems. Therefore, I suggest accepting this work with a revision and I hope the author goes through the paper another time to try to improve readability and fix any writing issues.

**Audience:**

The paper might be of interest to at least some audience at TMLR. While Reviewer x1vw pointed out the motivation for this work to be unclear, they found the extra result in Theorem 21 to give a justification for using fractional derivatives. Reviewers 6T9r and Reviewer x1vw noticed that the main results are for one-dimensional problems, and Reviewer 4UxH also wrote about the high-dimensional extension "... this is done in very limiting separable case", and this has been addressed by the additional results added to the paper during the discussion period.

**Claims And Evidence:**

The author did a very good job of making claims that are just, fair, and supported by evidence. The results are indeed novel and the proofs are rigorous. There is not much empirical evaluation in the paper, but the paper does not make any claims about the practical aspects, so everything is in line with the guidelines.